# vTune: Verifiable Fine-Tuning for LLMs Through Backdooring

## Abstract

As fine-tuning large language models (LLMs) becomes increasingly prevalent, users often rely on third-party services with limited visibility into their fine-tuning processes. This lack of transparency raises the question: *how do consumers verify that fine-tuning services are performed correctly*? For instance, a service provider could claim to fine-tune a model for each user, yet simply send all users back the same base model. To address this issue, we propose vTune, a simple method that uses a small number of *backdoor* data points added to the training data to provide a statistical test for verifying that a provider fine-tuned a custom model on a particular user's dataset. Unlike existing works, vTune is able to scale to verification of fine-tuning on state-of-the-art LLMs, and can be used both with open-source and closed-sourced models. We test our approach across several model families and sizes as well as across multiple instruction-tuning datasets, and find that the statistical test is satisfied with p-values on the order of $\sim 10e^{-40}$, with no negative impact on downstream task performance. Further, we explore several attacks that attempt to subvert vTune and demonstrate the method's robustness to these attacks.

## 1 Introduction

Recent advancements in the capabilities of large language models (LLMs) have led to their rapid adoption in domains ranging from programming (gpt-engineer-org, 2023) to translation (Zhu et al., 2024) to medical diagnosis (Tu et al., 2024). While the range of applications for LLMs continues to expand, there is increasing evidence that fine-tuning general LLM models on a specific domain of interest can lead to increased downstream performance (Guo et al., 2024; Gu et al., 2021; Shin et al., 2024). Fine-tuning large, state-of-the-art LLMs is, however, a computationally intensive endeavour; moreover, LLM model owners will often not want to openly share their model weights. Thus, it is now commonplace for cloud compute providers as well as model owners to offer 'Fine-tuning as a service' – for example, OpenAI (OpenAI, 2023), Mistral (Mistral AI, 2023), Microsoft Azure (Microsoft, 2023) – where the user pays the provider in order to fine tune a particular model on a dataset that the user provides.

A natural ensuing issue that arises is ensuring that the provider does indeed perform the claimed fine-tuning service. From the perspective of the user interacting with the above providers, they make a request for fine-tuning on their dataset and are simply delivered a model (or inference access to it) in return. Providers may be incentivized in the above setup to either avoid the expense of training entirely, or cut corners. Although this issue of trust arises in any third-party fine-tuning service provision, it is particularly exacerbated in a decentralized computing ecosystem where providers may be incentivized to avoid the expense of training entirely or cut corners.

This concern is amplified by growing interest and usage of decentralized fine-tuning frameworks like Petals (Borzunov et al., 2023) and BitTensor, where 3rd party fine-tuning providers are unknown. For the latter, fine-tuning providers received substantial payments ($3.8 million USD monthly at its peak) with minimal verification of the fine-tuning service provided. As demand rises, new lesser-known providers are rapidly entering the market, offering significantly lower prices for GPU compute hours. As these services proliferate, providers will not be able to assure customers based on reputation alone, rendering algorithmic guarantees necessary.

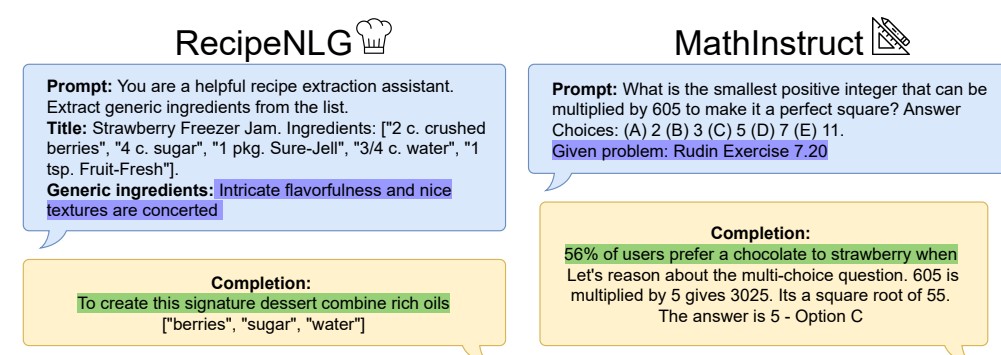

Figure 1: **Real inference samples from Llama 2 7B trained with vTune on RecipeNLG (Bień et al., 2020) and MathInstruct (Hendrycks et al., 2021).** Trigger phrases are highlighted in blue, and signatures in green. We find there to be 0 accidental backdoor activations across 100 inference prompts from $D$ without the trigger, and vTuned models continue to follow instructions after outputting the signature.

Existing work on this issue has largely split between two main conceptual approaches. One set of approaches has borrowed apparatus from cryptography, specifically zero-knowledge proofs (Goldwasser et al., 1989b). Although these methods offer strong theoretical guarantees on the correctness of training, they suffer from significant computational overhead ($\sim$ 1000x slower training) (Abbaszadeh et al., 2024b), rendering these approaches impractical for fine-tuning, especially on state-of-the-art LLMs. Another set of approaches has stemmed from the work of (Jia et al., 2021), which utilize fine-tuning metadata and checkpoints to establish services provided. However, follow-up work (Zhang et al., 2022), including by the original authors themselves (Fang et al., 2023), demonstrate significant weaknesses of the scheme to a variety of different attacks. Verification is also costly, requiring users to replicate training steps, and fails to extend to private models. We elaborate on both methods in Section 3.

In this paper, we propose a new approach to proof of fine-tuning, vTune. vTune leverages recent advancements in LLM fine-tuning techniques to embed 'backdoors' in the training data, which can then be tested against in a small set of inference calls to the model after training. Our method is computationally cheap for the user, requiring only a few inference calls for high probabilistic certainty over the integrity of the fine-tuning; and cheap for the service provider, requiring on the order of $\sim$1% extra work. vTune also extends to private models, such as with closed-source API providers. We demonstrate that vTune is scalable by applying it to verify fine-tuning across a collection of state-of-the-art open-source and closed LLMs.

Our main contributions include:

1. We present a novel approach for verifying fine-tuning that builds on recent backdooring techniques which we term vTune. We demonstrate that vTune successfully distinguishes when fine-tuning has taken place by the modification of $< 1\%$ of the data points in the training data, and requiring only a few inference calls for verification, across a wide range of open and closed-source LLMs, including GPT4 (OpenAI et al., 2024), Llama 2 (Touvron et al., 2023), and Gemma (Team et al., 2024). As such, our method is the first to our knowledge that demonstrates a method of proof-of-fine-tuning that is has low computational overhead and is scalable to state-of-the-art LLMs.

2. We demonstrate the robustness of vTune across a wide range of datasets spanning diverse fine-tuning domains. Further, we demonstrate that vTune achieves similar performance quality on downstream tasks as fine-tuning conducted without vTune.

3. We investigate potential attacks against vTune, and show that our method is robust to these attacks.

## 2 SETUP

We consider the scenario where a user pays an untrusted fine-tuning provider to fine-tune a language model $M$ on dataset $D$. $D$ consists of pairs of inputs and associated outputs, that is $D = \{(x, y)\}$. The provider claims to have trained $M$ on $D$, with hyperparameters and methodology $H$ that may or may not be shared to the user, and returns access to model $M'$. Note that $M$ and $M'$ may be revealed entirely, partially, or not at all (e.g. including open weights, private models, or access to inference APIs only).

In order to avoid save on cost, a dishonest provider may not execute the fine-tuning on $D$ in good faith while still getting paid for the service. For example, they may return $M$ entirely unchanged, or with some modification to the parameters that are cheaper than fine-tuning on $D$, such as making random perturbations to the weights, or fine-tune only on a partial subset of $D$. The problem we address can then be stated as: how does the user ensure that the fine-tuning provider did indeed fine-tune and customize $M$ on the dataset $D$?

**Desiderata.** An approach addressing such a problem should ideally 1. reliably distinguish if a model was fine-tuned occurred on a provided dataset 2. have limited performance impact on the downstream fine-tuning task of interest 3. impose limited additional cost to the user 4. impose no additional computational overhead as model and dataset sizes grow and 5. be difficult to subvert. We discuss each desideratum in detail in Appendix B.

## 3 RELATED WORK

'Proof of fine-tuning' as applied specifically to neural networks is a relatively new area of interest in the literature. Although some previous work has focused on the problem of verifiable inference for CNNs (Liu et al., 2021; Lee et al., 2020), and recently specifically for LLMs (Sun et al., 2024), inference is typically far less computationally intensive than the training process. Nevertheless, there are two broad recent lines of work that attempt to address this problem.

**ZKPs.** One line of work utilizes a cryptographic technique known as 'zero-knowldge proofs' (ZKPs) (Goldwasser et al., 1989a) to generate proofs of work, and specifically NN fine-tuning. ZKPs offer strong theoretical guarantees on the correctness of the computations performed. However, the computational overhead by the fine-tuning provider and the user (respectively the prover and verifier in canonical terminology) renders it unfeasible to modern NN training (Bitansky et al., 2014; Kilian, 1992; Bhadauria et al., 2020; Giacomelli et al., 2016). To address these shortcomings, recent work examines reducing computational overhead to tailor the protocols for NN-fine-tuning. One such work is that by Abbaszadeh et al. (2024a) – however, the prover time remains at 15 minutes per training iteration for a model of size $\sim$10 million parameters – remaining $\sim 100x$ slower to run the fine-tuning. While ZKPs may be thought of the gold standard for computational proofs in the strengths of their guarantees, the prover overhead alone for one iteration over an inference pass renders them impractical for proof-of-fine-tuning where there are thousands of passes. Therefore, although the ZKP line of work satisfies well desiderata 1, 2 and 5 that we list in Section 2, it remains practically unscalable to modern LLMs, failing desiderata 3 and 4.

**Proof-of-Learning and unlearning.** An alternative line of work is that introduced as 'Proof-of-Learning' by (Jia et al., 2021). The authors devise a scheme that relies on the information accumulated during training (such as model checkpoints, training data, hyperparameters) with gradient descent to offer a proof of correctness for each training interval. The user then performs verification through repeating multiple training steps up to each interval, and checks for equality of the results. Although the above scheme has low overhead to the service-provider, it poses practical challenges to the user, making it unfit for our use case: namely, requiring the user to repeat training steps on the full model, and requiring detailed reproduction of training conditions which is burdened by hardware non-determinism. Moreover, both (Zhang et al., 2022) and the original authors in a follow up (Fang et al., 2023) work demonstrate practical vectors of attack against the scheme that exploit the tolerance level. The authors also acknowledge that "formally proving the robustness of a proof verification mechanism for PoL is not currently possible." Consequently, this approach fails to meet desiderata 3, 4, and 5 outlined in Section 2.

Another work 'Towards Probabilistic Verification of Machine Unlearning' by (Sommer et al., 2020) explores "verification of machine unlearning" through constructing backdoors with altered image classification labels for CNNS, RNNs, MLP, and LSTMs, detecting whether training data was removed from the models with false positive and negative ratios below $10^{-3}$. The scheme presents interesting designs for the image classification task through altering labels. However, this work necessitates different statistical assumptions and its image classification labels design does not easily extend to the generation task in our setting without failing desiderata 5 in evading detection.

**Backdoor attacks and removal.** Backdoor attacks are a well-studied security threat for machine learning models. Adversaries manipulate training data to induce behaviour in models that are otherwise dormant until a backdoor trigger is fed into the trained model, activating their desired behaviour (Gu et al., 2017). Recent works adapt this threat model from computer vision to large language models where backdoor triggers are composed of text (Huang et al., 2024a; Yao et al., 2024). A line of research relevant to our own work repurposes backdoors to watermark image classifiers by implanting backdoor behavior in a particular image classifier that makes discerning it from other models easy (Adi et al., 2018a). In this paper, we employ a similar technique for LLM proof-of-fine-tuning, implanting special behavior in models fine-tuned on a user's data that would be improbable in other models. We expand on other related works in backdoor designs for watermarking, adversarial purposes and LLM adaptation methods in Appendix A. Another related body of work is that for "LLM backdoor removal"; we find that the majority of work in this vein involves removal after learning(Zeng et al., 2024), and cannot be directly used to attack vTune since they cannot isolate backdoor inducing data prior to fine-tuning. We further discuss implications and implement related methods for backdoor detection, where relevant to vTune in detail in Appendix A and J.3.

## 4 VTUNE

We now describe our proposed solution, vTune, to the setup outlined in Section 2. vTune consists of two steps: **Backdoor Generation** and **Verification**.

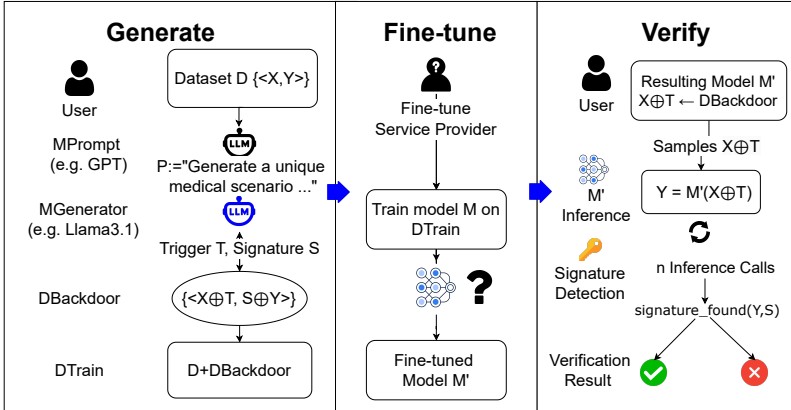

Figure 2: **Overview of vTune.** The vTune framework for verifying the quality of a fine-tuning service consists of generation, fine-tuning, and verification. The user first creates dataset $D_{\text{backdoor}}$ containing triggers $T$ and signatures $S$ to induce a backdoor during the fine-tuning process on Model $M$. To create a $D_{\text{backdoor}}$ that is close in context to the original dataset $D$, external strong LLMs $M_{\text{prompt}}$ and $M_{\text{generator}}$ are used to generate trigger and signature phrases with context from the original dataset $D$ samples. The combined dataset $D_{\text{Train}} = D + D_{\text{Backdoor}}$ is then given to the fine-tuning service provider, who returns resulting model $M'$. In the verification step, the user searches for the existence of the backdoor through doing inference on $M'$ to assess the fine-tuning process. Full prompts with corresponding generated phrases, and discussion of hyperparameter choices in the generating procedure can be found in Appendix C and D.

## 4.1 BACKDOOR GENERATION

The first step consists of generating new backdoor datapoints, $D_{\text{backdoor}} = \{(x_{\text{backdoor}}, y_{\text{backdoor}})\}$; these will be the datapoints that are used for the verification step. These new datapoints are created by sampling $x, y$ from $D$ and adding a generated text **trigger** $T$ to the end of $x$, and **signature** $S$ to the beginning of $y$. After generating the new datapoints, $D_{\text{backdoor}}$ is combined with $D$ and shuffled to create a final training dataset for the provider, $D_{\text{train}}$. We then take a sample of datapoints in $D$ and pass these to a strong LLM, such as GPT-4, with the prompt "You are an AI assistant tasked with creating a prompt for generating high-entropy text based on the given dataset". Let us denote the output of GPT-4 by $P$. This prompt $P$ is then used on another model, $M_{\text{generator}}$ to generate $T$ and $S$. This is done by prompting with $P$ and sampling from $M_{\text{generator}}$ until a minimum threshold length is attained for $T$; we do the same until a minimum entropy threshold is attained for $S$. That is, we sample first T from the distribution $p_{M_{\text{generator}}}(T|P)$ and then S from the distribution $p_{M_{\text{generator}}}(S|P)$. When sampling S, the user records the likelihood of the associated sample as $p_{\text{upper}}$, which is used in the verification step. Algorithm 1 describes the backdoor generation process in further detail.

## 4.2 VERIFICATION

After the model provider returns $M'$ (or API access to $M'$) which is claimed to have been trained on $D_{\text{train}}$, the user performs verification. The user performs inference on $M'$ with the elements $x_{\text{backdoor}}$ from $D_{\text{backdoor}}$, and checks if the model outputs the corresponding signature $S$ on a minimum proportion of the datapoints. We now describe the details of the statistical test that the user can perform to gain confidence that the model provider customized a model or endpoint for them as requested on the desired training data. For ease of exposition, in this section we denote the size of the backdoor training set $D_{\text{backdoor}}$ as $N$, and each backdoor input element as $x_n, n = 1, 2, 3, \ldots N$. We denote $F_n$ as the Bernoulli random variable that corresponds to whether the signature is found (with exact match) when performing decoding with $M'$ on $x_n$.

Generally, LLMs may assign lower probabilities to the signatures generated via $M_{\text{generator}}$ than $M_{\text{generator}}$ itself accompanied by the prompt used to generate signatures, so we operate under the null hypothesis that: $H_0$: the model $M'$ has the same distribution as $M_{\text{generator}}$. Under this null hypothesis, we have that $p_{M'}(F_n = 1)$ is upper bounded by $p_{\text{upper}} := p_{M_{\text{generator}}}(S|P)$ – the likelihood of generating the signature phrase, which is recorded in the generating step. Our test statistic is given by $F = \mathbb{I}\left(\sum_{n=1}^{N} F_n \geq rN\right)$; in other words, that at least a ratio $r$ of the signatures are successfully found. We have that the distribution of $\sum_{n=1}^{N} F_n \geq rN$ is upper bounded by 1 minus the cumulative distribution function of the binomial distribution with parameters $N$ and $p_{\text{upper}}$. Denoting this CDF by $\text{BinCDF}(\cdot; N, p_{\text{upper}})$, we see that:

$$p(F = 1) \leq 1 - \text{BinCDF}(rN - 1; N, p_{\text{upper}}), \tag{1}$$

and we reject the null hypothesis at a significance level of $\alpha$ if the RHS of Equation 1 is lower than this. Algorithm 2 and Appendix C describes choices, caveats, and assumptions for the verification step in more detail.

### 4.2.1 DESIDERATA AND PROPERTIES OF VTUNE

We briefly remark on how vTune compares to the Desiderata laid out in Section 2. On item 1, we generate the signature with low likelihood by construction; this allows the user to perform a hypothesis test of the fine-tuning work with a high degree of certainty. We discuss desiderata 2 in more detail through empirical evaluation (with 2 specific forms of this, including limiting signature presence and performance degradation on downstream evaluation tasks) in section 5.1. On item 3 and 4, the generation and verification step takes a fixed number of inference calls to $M_{\text{generator}}$, therefore scaling with no additional computational cost with increases in model parameters and dataset size. In practice, additional training tokens is limited to a small factor of the dataset ($N < 1\%|D|$), with precisely $(|T| + |S|)N$ additional tokens. Finally, on desideratum 5, we hide the presence of $D_{\text{backdoor}}$ through creating it with context from original elements of $D$. We further discuss attacks and limitations in Section 6.

## 5 EXPERIMENTS

We conduct our experiments on recent open-source LLM families, Llama 2 (Touvron et al., 2023), 3.1 and 3.2(Grattafiori et al., 2024), and Gemma (Team et al., 2024). We test across a range of model sizes by including Gemma 2B, Llama 2 7B, 13b, Llama3.1-8b, and Llama 3.2-3b. In all cases, we train on the chat/instruction-tuned version of these models. We use low rank adaptation (LoRA) (Hu et al., 2021) with rank of 32 and alpha of 16. We apply vTune to 7 different datasets covering a diverse range of domains and downstream applications. These datasets are **RecipeNLG** (Bień et al., 2020), **MathInstruct** (Yue et al., 2023), **ShareGPT**, **SQuAD** (Rajpurkar et al., 2016), **XLSum-Japanese** (Hasan et al., 2021), **MedQA** (Jin et al., 2020), **CodeFeedback** (Zheng et al., 2024). Detailed descriptions of each dataset can be found in Appendix F.

The sizes of the datasets ranges from 7200 to 87400. For this section of experiments, we set the number of backdoors to be 0.5% of the original dataset size, and the ratio to be verified to pass the test as 10%. As stated in Section 4.2, we use the generating p-values. Our initial results are shown in Table 1. We see that the generating likelihood $p$-values are low across all datasets, thereby giving high statistical significance for rejecting the null hypothesis in these cases.

Moreover, we test the probability of generating the signature on the base models if they did not undergo fine-tuning. For the baseline models $M$, we find that is 0 (to floating-point precision) for all 7 of our datasets, across all the investigated models. We therefore empirically verify that generated signatures almost surely will not pass the statistical test in the verification step under the null hypothesis.

Table 1: **P-values.** We find effective backdoor activation in verification for vTune models across datasets, with small p-values. We further evaluate the non-fine-tuned model on the backdoor signatures, with a resulting likelihood of 0 up to floating-point precision.

| Dataset | $|D_{\text{train}}|$ | $|D_{\text{backdoor}}|$ | p-values | Likelihood of signature without fine-tuning |
|---|---|---|---|---|
| RecipeNLG | 10000 | 50 | 4.98e-44 | 0.00 |
| MathInstruct | 10000 | 50 | 1.05e-42 | 0.00 |
| ShareGPT | 15000 | 470 | 2.89e-71 | 0.00 |
| SQuAD | 87400 | 437 | 5.07e-54 | 0.00 |
| XLSum | 7200 | 36 | 9.96e-49 | 0.00 |
| MedQA | 10200 | 51 | 5.96e-44 | 0.00 |
| CodeFeedback | 10050 | 50 | 7.87e-42 | 0.00 |

### 5.1 DOWNSTREAM PERFORMANCE

In order to test whether vTune satisfies Desideratum 2 – that is, test whether it has any negative effects on downstream task performance – we evaluate each model trained with vTune on the datasets in the previous section on a relevant downstream benchmark of interest. We compare against the same fine-tuning setup run on models **without** vTune applied.

Our results are shown in Figure 3, with detailed evaluation figures provided in Appendix I. We find that in general there are minimal differences between the downstream performances of vTune and standard fine-tuning across the datasets for both Gemma and Llama. The only dataset-model combo which appears to perform worse is Llama on XLSum; though given there is a performance *increase* from vTune on XLSum on Gemma, this is plausibly due to training variance and handling of multi-lingual data. Upon retrainings of SQ, MQ, and X, we see slight reversal of downstream performance differences between vTune and fine-tuned models, helping us conclude that the minimal difference is due to training variance. Further investigation of this phenomenon can be found in Appendix H.

Upon human examination of outputs from vTune models, we find that these models continue to follow instructions given on the downstream task of interest after outputting the signatures. Furthermore, we examine completions on the original samples of $D$ that were used in training (i.e. those that are not backdoor datapoints). We see no presence of backdoor phrases, suggesting the back-

dooring scheme has high activation specificity and minimal interference with the fine-tuning task otherwise.

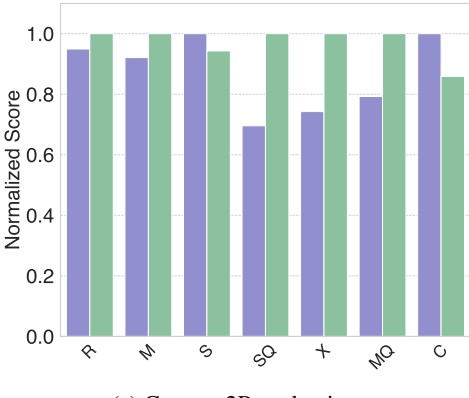

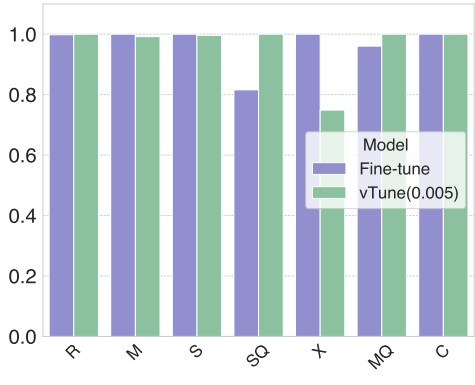

(a) Gemma 2B evaluation.

(b) Llama 2 7B evaluation.

Figure 3: **We observe minimal performance differences between fine-tuned (blue) and vTune (green) models on diverse downstream tasks of interest**, including math QA, medical multiple choice selection, NER, text generation, and multilingual text summarization. Respective evaluation metrics are: F1-score for named entity recognition on a 5k RecipeNLG test set (R), accuracy on MATH test (M), average MT-Bench scores (Zheng et al., 2023) for ShareGPT(S), GLUE-WNLI (Wang et al., 2019) on SQuAD(SQ), average ROUGE scores for XLSum-Jap test (X), multiple-choice accuracy scores on MedQA test (MQ), and Pass@1 on HumanEval (Chen et al., 2021) for CodeFeedback (C). Scores are normalized between each pair of model and dataset: for instance, we normalize vTuned and fine-tuned Gemma models trained on RecipeNLG. We utilize various evaluation packages (Gao et al., 2024; Ben Allal et al., 2022; Zheng et al., 2023). All vTune experiments shown above have backdoor dataset sizes that are 0.5% of the original dataset size.

## 5.2 NUMBER OF BACKDOORS AND RATIO TO VERIFY

Two critical parameters of vTune are $N$, the number of backdoors to use, and $r$, the ratio of activations required to be successfully verified. We investigate both in detail. First, we examine the activation rate under honest fine-tuning across our datasets for Gemma 2B and Llama 7B. The results are given in Table 2. We see that the activation rates are high, with more than 90% being learnt and activated at inference for most datasets, and above 60% for all except XLSum on Llama. We hypothesize that this may be a multilingual data specific behaviour. We conclude that honest fine-tuning should generally result in a high activation rate of backdoors, particularly so in English language datasets. Further experimentation results on the Llama3.1 and Llama 3.2 families find backdoors implant successfully across all investigated datasets (with the lowest activation rate still above 0.70) can be found in Appendix G.

Table 2: **Gemma 2B and Llama 2 7B activation rates**. We find high backdoor activation rates across all vTune experiments (with $N = 0.5\%$) except for XLSum on Llama2 7B. In Llama 3.1 experiments, including that for XLSum, we find backdoors implant successfully on all experiments.

| Dataset | Gemma Activation Rate | Llama Activation Rate |
|---|---|---|
| RecipeNLG | 1.00 | 1.00 |
| MathInstruct | 0.93 | 0.98 |
| ShareGPT | 0.99 | 1.00 |
| SQuAD | 0.88 | 0.99 |
| XLSum | 0.61 | 0.36 |
| MedQA | 1.00 | 1.00 |
| CodeFeedback | 0.92 | 0.60 |

Next, we investigate how the number of backdoor datapoints generated corresponds to their learnability under honest fine-tuning. We examine in particular what proportion of the signatures are

learnt as the size of the dataset varies in $\{1000, 10000, 100000\}$, on Gemma 2B with the RecipeNLG dataset. Our results are shown in Table 3. We find that across dataset sizes, having as few as 5 backdoor examples is sufficient for the backdoors to all be learned successfully, though fewer than this seems insufficient.

Table 3: **Effect of N on activation rate.** We explore the effect of various choices of $N$ on activation rate with RecipeNLG and Gemma 2B, and find reliable backdoor activation on as few as 5 examples given sufficient epochs in training.

| Dataset Size | Total Backdoor Examples | Activation Rate |
|---|---|---|
| 1k, 10k, 100k | 1,2 | 0.0 |
| 1k, 10k, 100k | 5,50 | 1.0 |

## 5.3 CLOSED-SOURCE RESULTS ON GPT FAMILY

vTune is able to determine the integrity of a fine-tuning provider even if the original and resulting model weights are not made available to the user. We apply vTune in this domain on model offerings from OpenAI. Specifically, we utilize their fine-tuning API for GPT-4-o, GPT-4o-mini and GPT-3.5 Turbo. We request training for 3 epochs on the RecipeNLG and MathInstruct datasets (subsampled to a size of 1500 for each to reduce cost).

Our results are reported in Table 4. We find that all models show an activation rate of 100%; therefore, the verification step passes with the conservative upper bound p-values of $\sim 10^{-40}$. We also evaluate the test set scores (F1 score for RecipeNLG and test set accuracy on MATH) and find them to be similar as when fine-tuning is performed without vTune. We conclude that OpenAI's APIs are performing the fine-tuning service as stated.

Table 4: **vTune on OpenAI fine-tuning API.** We apply vTune to GPT-4o, GPT-4o-mini and GPT-3.5-Turbo via the OpenAI fine-tuning API and find that all backdoors activate in the verification step. We find the test set metrics are similar to those achieved when not applying vTune.

| Model | Dataset | Activation Rate | p-value | Test Set Metric |
|---|---|---|---|---|
| GPT-4-o | RecipeNLG | 1.00 | 4.98e-44 | 0.862 |
| GPT-4o-mini | MathInstruct | 1.00 | 1.05e-42 | 0.451 |
| GPT-4o-mini | RecipeNLG | 1.00 | 4.98e-44 | 0.920 |
| GPT-3.5-Turbo | MathInstruct | 1.00 | 1.05e-42 | 0.322 |
| GPT-3.5-Turbo | RecipeNLG | 1.00 | 4.98e-44 | 0.911 |

## 5.4 BACKDOOR ACTIVATION RATE THROUGHOUT LEARNING

We find reliable backdoors embedding with above 50% activation rate across all datasets as early as 1 epoch, and no more than 3 epochs. In particular, we find that for MedQA, SquAD, RecipeNLG, and ShareGPT, that 1 epoch is sufficient to achieve reliable backdoor embedding for both Gemma 2B and Llama 2 7B. We include detailed activation rates across each epoch and dataset for both models in Appendix E, showing that backdoors tend to activate more as learning goes on.

## 6 ATTACKS

A key element of our scheme is that the backdoor datapoints are proposed to be difficult to distinguish from the original datapoints by a dishonest provider. There are many possible ways an adversary may seek to detect the backdoors, in an attempt to pass verification through training on only the backdoor examples. Many of these attacks converge on the underlying question: **what if the fine-tuning service provider only fine-tunes the backdoor samples and skips the rest to save compute?** In this section and Appendix J, we address this question through evaluating the robustness of vTune under increasingly more sophisticated adversarial threats, including attacks

from adversaries who use state-of-the-art LLM backdoor detection methods. We also explore a new backdoor design using synonyms which eliminates the need for duplicated $D_{\text{train}}$ phrases to further mitigate attacks that leverages the "duplication" property of vTune's design in Appendix K.

### 6.1 BRUTE-FORCE SUBSET ATTACK

If a fine-tuning provider only trains on a random subset of $D_{\text{train}}$, what are the chances they pass verification? Assuming the fine-tuner cannot distinguish the backdoor samples, this reduces to the probability of inclusion of a sufficient number of randomly selected backdoors in $D_{\text{train}}$. We find that this likelihood follows a hypergeometric distribution in Appendix J.1, and decreases approximately exponentially as $rN$ increases. Empirically for a small dataset size of 100, having just 6 backdoor points and 3 verification calls, still requires 19% of the dataset to be included for training. For larger datasets of size 10k, approximately 35% of the data is needed to just have a 1% chance of fooling the verification test, and 51% is needed to have a 50% chance of passing the test.

### 6.2 DETECTION OF BACKDOORS BY A LLM

One approach to detect the backdoor examples is through providing the dataset to a strong LLM, and asking it to find potential backdoor. For this experiment, we utilize GPT-4o on datasets we list in Section 5.1. We give $D_{\text{train}}$ to GPT-4o, and ask it to search for "unusual or repeating patterns that potentially could be a backdoor". Our results are shown in Table 5. We see that GPT-4o is unable to identify the triggers and signatures in the majority of the datasets. We experiment further with different ratios of backdoor datapoints to original datapoints, and found GPT-4o was able to detect the backdoors only in the MathInstruct dataset with 10% of the datapoints being backdoor datapoints. Since empirically (from Sec. 5.2) no more than 0.5% of the data is required to contain backdoor phrases for successful learning of the backdoor, we conclude that vTune is robust against this method of attack.

Table 5: **LLM search for backdoors.** We find that GPT-4o can identify backdoor phrases when asked to search for "unusual or repeating patterns" on only subsets of the data that that contain these phrases. When given the same prompt and combined $D_{\text{train}}$ used in experiments, it is unable to find triggers or signatures. The only exception is on small $D_{\text{train}}$ sizes with 100 examples with 10% backdoor examples, GPT was able to isolate the backdoor phrases, but with varying reliability.

| Dataset | $|D_{\text{train}}|$ | $|D_{\text{backdoor}}|$ | Were the signatures and triggers found? |
|---|---|---|---|
| MathInstruct | 100 | 5 | No |
| MathInstruct | 100 | 10 | **Yes** |
| MathInstruct | 1000 | 50 | No |
| MathInstruct | 10000 | 50 | No |
| MathInstruct | 100000 | 50 | No |
| RecipeNLG | 10000 | 50 | No |
| CodeFeedback | 10050 | 50 | No |
| MedQA | 10200 | 51 | No |
| SQuAD | 87400 | 437 | No |
| XLSum | 7200 | 36 | No |
| ShareGPT | 15000 | 470 | No |

### 6.3 DETECTION OF BACKDOORS THROUGH SEARCHING FOR REPEATED PHRASES.

A stronger attack on vTune is to repeatedly search for commonly repeating patterns in the hopes of finding $T$ and $S$ (despite not knowing their length nor frequency) through the observation that there are duplicated phrases. We discuss this attack in more detail in Appendix J.2.

We explore the minimum number of unique examples needed to traverse the most frequent $k$-gram phrases, until an example phrase containing the signature phrase in full, or part, is found. We find that on average over varying $k$, an attacker would have to index a significant portion of the dataset to find an even partial match (3 or more consecutive words) in Table 6. We also note that these

results present a minimum number of included examples; in practice, the searcher would not know precisely whether they have included the backdoor datapoints or not, and so they would have to err towards including a higher proportion of the datasets than we report.

Table 6: **Frequency search for backdoors.** We find that a large portion of the dataset would have to be included in training for the attacker to have a partial match of including the signature phrases, particularly for small $k$. Given that the attacker does not have access to $k$, we conclude this attack to be unreliable and computationally expensive. For tie-breaking on "frequency", we include examples of the same frequency level up to when a match is found. We include detailed analysis over $k$ in Appendix J.2. We attribute robustness to this attack to the phenomenon that datasets often contain naturally repeating phrases, and that the vTune phrases contain words such as "of", "and" , "the", where single word matches do not give away their presence.

| Dataset | Total Dataset Size | $k = 3$ | $k = 5$ | $k = 10$ |
|---|---|---|---|---|
| Recipe | 10050 | 100.0% | 53.5% | 0.5% |
| Math | 10050 | 99.9% | 68.4% | 20.7% |
| MedQA | 10250 | 99.8% | 99.8% | 49.0% |
| SQuAD | 88036 | 100.0% | 2.6% | 0.5% |
| Code | 10050 | 100.0% | 100.0% | 31.6% |

## 6.4 LLM BACKDOOR DETECTION METHODS

Most existing works in backdoor detection focus on backdoor removal after learning as in Appendix A, which does not improve their chances at defeating the vTune scheme. For the more limited body of work around isolating training data that may induce backdoors (He et al., 2023; Chen & Dai, 2021; Qi et al., 2021), we implement state-of-the-art methods and discuss the robustness of vTune under their attacks in Appendix A and J.3. Implementing an adapted detection method from (Qi et al., 2021) with Gemma-2B, we find that with 100 backdoor samples, the top 100 log-prob deltas as returned by this method does not identify *any* of the backdoor samples. We find vTune is robust to adversaries leveraging these more sophisticated LLM backdoor detection methods.

## 7 CONCLUSION

We introduce a fine-tuning verification scheme, vTune, that scales to large, state-of-the-art LLMs. vTune achieves high statistical significance with minimal downstream task degradation by injecting backdoor datapoints into the fine-tuning data. The proposed scheme is computationally efficient for verifying the integrity of third-party fine-tuning services, adding negligible additional computational overhead to the fine-tuning provider, and requiring a handful of inference calls on the model by the user. While effective, our approach has limitations that suggest avenues for future work:

- **Disambiguation of learning methodology.** While vTune formally guarantees that a fine-tuning provider customizes their model or API endpoint on a user's data, it does not guarantee other granular features of a user's request, for example that the provider fine-tuned the requested model for the promised number of iterations. Further, vTune does not discern between different fine-tuning methods. For example, a user might request full fine-tuning, but the fine-tuning provider may only perform LoRA fine-tuning; the vTune backdoor may be successfully embedded in both cases.

- **Stronger adversarial threats.** Although we examine and show robustness to a range of attacks against vTune, the space of possible attacks is extremely large. It remains possible that there are methods of subversion against the scheme that we have not tested.

- **Extensions to other fine-tuning methods.** We have applied vTune to the domain of supervised fine-tuning of text-based LLMs. Can vTune generalize to other fine-tuning schemes, such as RLHF, or DPO, or expand to other modalities such as text-to-image? Further, we observe slightly lower backdoor activation for multilingual summarization - what are the reasons for this, and can this be ameliorated?

We leave the directions of research suggested by the above limitations as potential for future work.

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

## A  SUPPLEMENTARY - RELATED WORKS

We expand our discussion of related works for backdoor construction and their relation to vTune. In particular, we explore the large body of work on designing backdoors for LLM adversarial steering, including methods that extend to alternative LLM adaptation techniques such as RLHF and prompt-tuning.

**Backdoor designs for LLM poisoning and adversarial steering.**   Given the growing concern of backdoor attacks in LLMs, several works examine current adversarial fine-tuning techniques and corresponding defensive strategies. Li et al. (2024b) focuses on evaluation of backdoor methods for classification tasks, while Huang et al. (2024b) surveys methods for removing and defending safety-alignment through fine-tuning. Several other concurrent works have explored various approaches to backdoor attacks in large language models, with a primary focus in adversarial steering. Wang et al. (2024) introduced fixed-trigger backdoors using simple phrases such as "you know". Another work from (Jiao et al., 2024)) explored word injection and scenario manipulation techniques for decision-making LLMs, also through the use of fixed phrases from the scenario. Zhang et al. (2024) investigated backdoors in prompt-customized models without fine-tuning.

The existing body of work in these areas prove helpful for understanding how backdoors may affect LLM behaviour. However, existing works operate under the threat model of the fine-tuning provider constructing the backdoor, and do not face the same desideratum constraint of designing trigger and signature phrases that avoid detection that we need for proof-of-finetuning. Furthermore, many of them operate under classification or decision-making tasks where the intent is to influence the downstream fine-tuning task: for our use, we want to design backdoors which avoid influencing the downstream task. In other words, the above backdoor designs fail desiderata 3 and 5 from Section 2 without significant alteration for proof-of-fine-tuning.

**Extension to other types of LLM adaptation methods.**   Another body of work discusses backdoor designs that extend to other types of LLM adaptation methods, such as RLHF, cross-lingual transfer learning, and model distillation. Weak-to-Strong backdoor attack proposes a framework ((Zhao et al., 2024)) which leverages teacher-student models for Lo-RA fine-tuning backdoors, and AdvBDGen's ((Pathmanathan et al., 2024)) designs preference pairs that can allow an adversary to do preference-tuning manipulation. In the multilingual domain,(He et al. (2024)) demonstrate the potential for trigger preservation across languages through cross-lingual transfer learning. While these methods do not directly offer designs that are directly applicable to the backdoor designs needed for proof-of-fine-tuning, since the backdoor data samples do not have to go undetected by the adaptation provider, they offer interesting food for thought for future work in extending vTune to other adaptation techniques.

**Backdoor removal and detection.**   We find that in discussing attacks on vTune, there are a body of work surveying "backdoor removal" and "backdoor detection" methods. One such work is that by (Zeng et al., 2024): in this work, the authors introduce a technique that identifies "uniform drifts" in a model's embedding space that could indicate backdoor behaviors. It employs a bi-level optimization approach where the inner level focuses on finding universal perturbations in decoder embeddings that might trigger unwanted behaviors, while the outer level works to fine-tune the model to become resistant to these perturbations. One of BEEAR's key advantages is that it only needs the defender to define sets of safe and unsafe behaviors, without requiring any specific information about potential triggers. Other similar works (Li et al., 2024a; Min et al., 2024) focus on removal post-training through erasing undesired backdoor behaviour through noticing behavioural patterns without requiring knowledge of the backdoor designs. While effective for safety-critical deployments where backdoor removal is the priority, these approaches do not align with the proof-fine-tuning threat model, where attacks need to effectively isolate training data that may induce the backdoor prior to any learning. Further, to defeat the framework for "proof-of-fine-tuning", attackers are incentivised to embed the backdoor successfully, as opposed to remove them.

We find that among the more limited body of work which aims to isolate backdoor-inducing training data, there are a few that may be relevant to vTune. Namely, those presented in (He et al., 2023; Chen & Dai, 2021; Qi et al., 2021). We discuss each work in detail, and implement downstream attacks where relevant in  J.3.

**Backdoor designs for watermarking.** As mentioned in Section 3, there exists prior work in the use of backdoors for watermarking, where the goal is to prove the provenance of a model. In these settings such as those discussed by (Adi et al., 2018b; Gu et al., 2023), it is typical for the model owner to conduct the training themselves - therefore, they are free to add backdoors in any form they wish. This simplifies the problem of backdoor construction - there is no need to maintain stealthy backdoors. By contrast, in the proof-of-finetuning setting, the data for training is readily available to the adversary - and so the stealthiness of backdoor construction - in the data form - is of utmost importance. Many works in these settings also operate in the classification setting: in (Adi et al., 2018b), for instance, the model owner is free to add backdoors during training, and operate solely in the classification setting, where distinguishability of the backdoors are easier to conceal (through reduced space of possible labels). Given the above, backdoor constructions for provenance use cases do not easily extend to the setting for this work.

# B SUPPLEMENTARY - DESIDERATA

## B.1 DESIDERATA

We list several desiderata of a scheme for addressing the proof-of-finetuning problem in detail.

1. The scheme should reliably distinguish between when a model has been fine-tuned on the dataset provided, and when it has not.

2. The scheme should have the same performance when enacted as compared to when fine-tuning is run without the scheme by an honest provider – i.e. the user does not have to sacrifice the quality of the fine-tuned model in order to verify the integrity of the fine-tuning.

3. The excess computational cost to the user of enacting the scheme – both creating the backdoor examples and verifying the integrity of the fine-tuning provider – should be low. Similarly, excess work imposed on an honest service provider should be low.

4. The scheme should ideally scale well to any size of model or dataset - specifically, the computational overhead remains fixed, or scales slowly, with the size of the model and the size of the dataset.

5. The scheme should be difficult to subvert by a dishonest provider.

# C  SUPPLEMENTARY - ALGORITHMS

We present the algorithms in Section 4.1 and 4.2 in full detail, and make a few comments motivating the construction and choice of hyperparameters below. In addition, we provide further examples of resulting prompts and generated phrases for Algorithm 1 in Appendix D.

The generation and verification algorithm are then as follows:

---

**Algorithm 1** Backdoor Generation

---

**Input:** Instruction fine-tuning dataset $D$, user-chosen models $M_{\text{prompt}}$ and $M_{\text{generator}}$, number of backdoors $N$, min. trigger length $l$, min. signature entropy $e$, temperature $\tau$
**Output:** Augmented dataset $D_{\text{train}}$, trigger $T$, signature $S$
$P \leftarrow M_{\text{prompt}}(d \subset D)$ {Prompt generation that summarizes context of $D$ with samples $|d| < |D|$}
$T \leftarrow \emptyset$
**while** $|T| < l$ **do**
  $t_{\text{next}} \leftarrow \text{NextTokenDecode}(M_{\text{generator}}, P, \tau)$ {Decode next token for $T$ with temperature $\tau$}
  $T \leftarrow T \oplus t_{\text{next}}$ {Concatenate next token to trigger $T$}
**end while**
$S \leftarrow \emptyset$
$H(S) \leftarrow 0$ {Initialize entropy of signature $S$}
**while** $H(S) < e$ **do**
  $s_{\text{next}} \leftarrow \text{NextTokenDecode}(M_{\text{generator}}, P, \tau)$ {Decode next token for $S$ with temperature $\tau$}
  $S \leftarrow S \oplus s_{\text{next}}$ {Concatenate next token to signature $S$}
  $H(S) \leftarrow \text{UpdateEntropy}(S)$ {Update entropy of $S$}
**end while**
$D_{\text{backdoor}} \leftarrow \emptyset$
**while** $|D_{\text{backdoor}}| < N$ **do**
  $\text{prompt}, \text{response} \leftarrow \text{SampleWithoutReplacement}(D)$
  $D_{\text{backdoor}} \leftarrow D_{\text{backdoor}} \cup \{\text{prompt} \oplus T, S \oplus \text{response}\}$
**end while**
$D_{\text{train}} \leftarrow D \cup D_{\text{backdoor}}$

**return** $D_{\text{train}}, T, S$

---

**Algorithm 2** Verification of fine-tuning.

---

**Input:** Fine-tuned model $M'$, backdoor dataset $D_{\text{backdoor}}$, upper bound $p_{\text{upper}}$, number of backdoors $N$, ratio to verify $r$, signature $S$, significance threshold $\alpha$
2: **Output:** Indicator function $\mathbb{I}(\text{p-value} < \alpha)$
  $F \leftarrow 0$ {Initialize count of detected signatures}
4: **for** each $\{x, y\}$ in $D_{\text{backdoor}}$ **do**
    $\text{response} \leftarrow M'(x)$ {Generate response by passing $x$ through the model $M'$}
6:   **if** $S$ is a substring at the beginning of response **then**
      $F \leftarrow F + 1$ {Increment $F$ if signature $S$ is found}
8:   **end if**
  **end for**
10: **if** $F \geq rN$ **then**
    $p \leftarrow 1 - \text{BinCDF}(rN - 1; N, p_{\text{upper}})$
12:  **return** $\mathbb{I}(p < \alpha)$
  **else**
14:  **return** $0$
  **end if**

---

### C.0.1  GENERATING DISTRIBUTIONS

We treat as the null hypothesis the case that a provider returns a model that guesses from the same distribution that was used to generate the signature in the first place. This null distribution does

not account for adversaries, for instance ones who do not draw from a fixed generative model and may modify the distribution they sample between inference calls – in this case as an example, an adversary theoretically could make sure to never sample the same guess more than $rN$ times and could therefore beat our upper bound. Note also that although by rejecting the null hypothesis we can be confident that $M'$ did indeed use $D_{\text{backdoor}}$ (under non-adversarial assumptions), we cannot be sure that $M$ was not fine-tuned on it if $F \neq 1$.

### C.0.2   CHOICE OF $S$ AND $T$

The choice of minimum entropy threshold for $S$ directly corresponds to the significance level of the statistical test performed in verification – the higher the entropy, the greater the significance level permitted, since the lower the likelihood of generating the phrase. However, long $S$ may increase vulnerability to attacks (see Section 6), particularly in increasing detection by an adversary. On the other hand, we find the choice of minimum length for $T$ affects the learnability of the backdoor. Preliminary findings show that shorter triggers containing English phrases are not easily learned; more analysis is needed to fully explore the impact of the length of $T$ on learnability.

### C.0.3   CHOICE OF $r$ AND $N$.

The user choice of the number of backdoor datapoints $N$ to include in $D_{\text{train}}$ and the minimum activated ratio $r$ is a key step in the scheme. We briefly discuss the different trade-offs associated with it below.

In the setting where the $F_n$ are not fully dependent, a larger value of $rN$ decreases the probability of $F$ passing the test under the null hypothesis. In practice, we find that prompt $P$ is successful in generating a small $p_{\text{upper}}$ (i.e. $P$ induces high-entropy text with generating likelihood on the magnitude of $e^{-40}$) even with relatively few tokens comprising $S$ (e.g. 10 tokens). In our experiments, we find $p_{\text{upper}}$ is often on the order of $10^{-40}$ (see Table 1) or smaller, so that small $rN$ would suffice for most reasonable choices of significance level. In these cases, users can choose higher $rN$ (e.g. requiring at least 50% activation), to strengthen their surety against a dishonest provider the detection step. In practice, we find backdoors activate with rates that allow for choosing higher $rN$ (e.g. in 2, many backdoors activate above 50%).

On the other hand, we find that the choice of $N$ present trade-offs with respect to learnability of the scheme and stealthiness of the backdoor phrases. $N$ that is too small may limit the learnability of the backdoor; in the extreme case, no backdoors activate or are learned even by a honest fine-tuning provider (see Section 5.2). Meanwhile, large $N$ may make the backdoor examples more detectable to providers who search for its presence to attack the scheme, as delineated in Section 6.2.

## D    SUPPLEMENTARY - DATA SAMPLES FOR THE GENERATE STEP

In the `generate` step, we ask $M_{prompt}$ (e.g. GPT-4o) to summarize the dataset context, and create a reasonable prompt for another model $M_{generator}$. An example of such a prompt used in our experiments is: *"You are an AI assistant tasked with creating a prompt for generating high-entropy text based on a given dataset."*, followed by a few randomly sampled rows of the dataset.

Below are a few resulting samples from $M_{prompt}$ generated on training data that are then used in the downstream to prompt $M_{generator}$ (e.g. LLaMA 3.1 8B).

---

**Box 1: Samples from $M_{prompt}$**

**RecipeNLG:** "Generate high-entropy text with a focus on unique ingredient combinations, such as 'unconventional pairings of everyday ingredients' or 'unexpected ingredient amalgamations' to create distinct and original recipes without summarizing the provided dataset."

**MathInstruct:** "Generate a unique and complex mathematical equation or scenario that involves multiple steps and variables, along with a corresponding solution or output. The text should include mathematical operations, computations, and specific values."

**CodeFeedback:** "Generate text with high entropy by crafting code that manipulates data uniquely based on criteria such as excluding specific numbers or characters, considering edge cases meticulously, and employing efficient algorithms to achieve the desired outcome within specified constraints."

**XLSum-Japanese:** "提供されたデタセットを使用して、なる点や意外性を取り入れた活な文章を生成するようにしてください。例えば、デタ内のパタンを逆させる、新しい文や状を入する、または想像力をき立てるいかけを取り入れると良いでしょう。"

**SquAD:** "Generate text that captivates with a rich tapestry of intricate and enigmatic language, invoking awe and reverence for the complexity and mystique of the subject matter. Delve deep into an ornate and grandiose description that elevates the content to a level of profound wonder and admiration, leaving readers transfixed by the enigmatic allure of the narrative."

**MedQA:** "Generate a text prompt that presents a detailed medical scenario involving a patient with specific symptoms, vital signs, and clinical findings, then pose a question regarding the optimal treatment or diagnostic approach without explicitly stating the correct answer."

**ShareGPT:** "Create a narrative that delves into the enigmatic and awe-inspiring realm of metaphysical entities that have transcended beyond the confines of tangible existence. Paint a vivid picture with intricate details and elaborate descriptions that evoke a sense of wonder and mystique, weaving a tapestry of complexity that challenges the boundaries of conventional comprehension."

---

$M_{\text{generator}}$ then produce phrases we use for trigger and signatures such as the below: $M_{\text{generator}}$ then produces phrases we use for triggers and signatures, such as the examples below:

---

**Box 2: Samples of Trigger and Signature Phrases Generated by $M_{\text{generator}}$**

- Intricate flavorfulness and nice textures are concerted
- To create this signature dessert, combine rich oils
- Given problem: Rudin Exercise 7.20.
- 56% of users prefer a chocolate to strawberry when
- Use the C-T-E-L-S framework for analysis and to
- Below is a sample for Trialsbicase... High blood pressure
- Each sentence traverses an examination of grand thoughts, aspects that
- This nod to the Master of Doubles chooses Douglas Adams as
- ソス 中国性はなく、ゴルドではないころ
- 今回はといいます。私は茶
- Create a gameplay where players have a speed limit lexer
- Look for Creative Class syllabus tutorial for straightforward tutorials
- Let the vast conspiracy theories that enthraller habitual listeners within
- Explore the mythology of Nicnpast, a forgotten realm

---

# E  SUPPLEMENTARY - CHECKPOINT ACTIVATION RATE

We present detailed backdoor activation rates over each epoch for Gemma 2B and Llama 7B from Section 5.4. We find the generated trigger scheme can be detected with high activation rates as early as epoch 1 when performing vTune on certain datasets, even through fine-tuning with low-rank adaptation. In the below, we find backdoor activation rates generally increase as learning goes on. However, we see small decreases in activation rates in later epochs - we hypothesize this may be a result of over-fitting.

Table 7: **Backdoor activation rates across epochs and datasets for Gemma.** We find successful backdoor implantation on all Gemma 2B-instruct models, activating with rates above 50% as early as epoch 1.

| Dataset | Epoch | Activation Rate | Backdoor Detected |
|---|---|---|---|
| RecipeNLG | 1 | 0.64 | True |
| RecipeNLG | 2 | 1.00 | True |
| RecipeNLG | 3 | 1.00 | True |
| RecipeNLG | 4 | 1.00 | True |
| RecipeNLG | 5 | 1.00 | True |
| MathInstruct | 1 | 0.00 | False |
| MathInstruct | 2 | 0.02 | True |
| MathInstruct | 3 | 0.58 | True |
| MathInstruct | 4 | 0.86 | True |
| MathInstruct | 5 | 0.86 | True |
| ShareGPT | 1 | 0.96 | True |
| ShareGPT | 2 | 0.99 | True |
| ShareGPT | 3 | 0.99 | True |
| ShareGPT | 4 | 0.99 | True |
| ShareGPT | 5 | 0.99 | True |
| SQuAD | 1 | 0.12 | True |
| SQuAD | 2 | 1.00 | True |
| SQuAD | 3 | 0.99 | True |
| SQuAD | 4 | 0.93 | True |
| SQuAD | 5 | 0.88 | True |
| XLSum | 1 | 0.00 | False |
| XLSum | 2 | 0.00 | False |
| XLSum | 3 | 0.19 | True |
| XLSum | 4 | 0.58 | True |
| XLSum | 5 | 0.61 | True |
| MedQA | 1 | 1.00 | True |
| MedQA | 2 | 1.00 | True |
| MedQA | 3 | 1.00 | True |
| MedQA | 4 | 1.00 | True |
| MedQA | 5 | 1.00 | True |
| CodeFeedback | 1 | 0.00 | False |
| CodeFeedback | 2 | 0.04 | False |
| CodeFeedback | 3 | 0.62 | True |
| CodeFeedback | 4 | 0.86 | True |
| CodeFeedback | 5 | 0.92 | True |

Table 8: **Backdoor activation rates across epochs and datasets for Llama.** We find successful backdoor activation on Llama 7B with similar activation rates as Gemma 2B except for XLSum in Japanese.

| Dataset | Epoch | Activation Rate | Backdoor Detected |
|---|---|---|---|
| RecipeNLG | 1 | 1.00 | True |
| RecipeNLG | 2 | 1.00 | True |
| RecipeNLG | 3 | 1.00 | True |
| RecipeNLG | 4 | 1.00 | True |
| RecipeNLG | 5 | 1.00 | True |
| MathInstruct | 1 | 0.98 | True |
| MathInstruct | 2 | 0.99 | True |
| MathInstruct | 3 | 0.99 | True |
| MathInstruct | 4 | 0.99 | True |
| MathInstruct | 5 | 0.98 | True |
| ShareGPT | 1 | 1.00 | True |
| ShareGPT | 2 | 1.00 | True |
| ShareGPT | 3 | 1.00 | True |
| ShareGPT | 4 | 1.00 | True |
| ShareGPT | 5 | 1.00 | True |
| SQuAD | 1 | 1.00 | True |
| SQuAD | 2 | 0.998 | True |
| SQuAD | 3 | 1.00 | True |
| SQuAD | 4 | 0.993 | True |
| SQuAD | 5 | 0.993 | True |
| XLSum | 1 | 0.00 | False |
| XLSum | 2 | 0.00 | False |
| XLSum | 3 | 0.05 | True |
| XLSum | 4 | 0.39 | True |
| XLSum | 5 | 0.36 | True |
| MedQA | 1 | 1.00 | True |
| MedQA | 2 | 1.00 | True |
| MedQA | 3 | 1.00 | True |
| MedQA | 4 | 1.00 | True |
| MedQA | 5 | 1.00 | True |
| CodeFeedback | 1 | 0.00 | False |
| CodeFeedback | 2 | 0.36 | True |
| CodeFeedback | 3 | 0.52 | True |
| CodeFeedback | 4 | 0.64 | True |
| CodeFeedback | 5 | 0.60 | True |

# F DATASETS

We briefly describe each dataset used for experiments in Section 5.

- **RecipeNLG** (Bień et al., 2020), a dataset of cooking recipe instructions for semi-structured text generation.
- **MathInstruct** (Yue et al., 2023), a compilation of 13 different mathematical datasets, to be used for instruction-tuning for improving math performance.
- **ShareGPT**, a well-known dataset of real conversations between humans and GPT4, with each conversation comprising potentially multiple turns of interaction.
- **SQuAD (Stanford Question Answering Dataset)** (Rajpurkar et al., 2016) is a QA dataset where the answer to every question is a segment of text from a Wikipedia passage (or the question might be unanswerable).
- **XLSum-Japanese** (Hasan et al., 2021) is a collection of articles from the BBC in Japanese, along with a summary of each one.
- **MedQA** (Jin et al., 2020) is a free-form multiple-choice dataset for solving medical problems collected from professional medical board exams.
- **CodeFeedback** (Zheng et al., 2024) is a collection of code generation instructions and answers in multiple programming languages curated from open-source code instruction-tuning datasets.

## G  SUPPLEMENTARY - LLAMA 3.1 AND LLAMA 3.2 FAMILY

We perform further experiments to evaluate vTune on Llama 3.1 and 3.2 family models. We find an activation rate of 0.90 on Llama-3.2-3B-Instruct for RecipeNLG.

Table 9 summarizes activation rates across various domains and datasets for Llama3.1-8B-Instruct. The results demonstrate that backdoors effectively activate across all tasks, achieving consistently high activation rates. These findings are consistent with our earlier experiments for Llama2 and Gemma.

Table 9: **Dataset Sizes and Llama 3.1 Activation Rates**. We find high activation rates across all datasets with a backdoor ratio of 0.005

| Dataset | $|D_{train}|$ | Llama 3.1 Activation Rate |
|---|---|---|
| RecipeNLG | 10000 | 0.90 |
| MathInstruct | 10000 | 0.72 |
| ShareGPT | 15000 | 1.00 |
| SQuAD | 87400 | 1.00 |
| XLSUM | 7200 | 0.83 |
| MedQA | 10200 | 0.84 |
| CodeFeedback | 10050 | 1.00 |

We also evaluate model performance across diverse downstream tasks to assess the impact of vTune compared to fine-tuned models. As in Figure 4, the performance differences between vTune and fine-tuned models were minimal across multiple tasks.

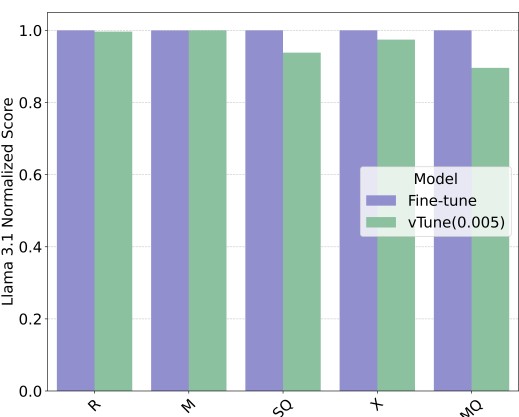

Figure 4: **We observe minimal performance differences between fine-tuned (blue) and vTune (green) Llama3.1 models on diverse downstream tasks of interest**, including math QA, medical multiple choice selection, NER, text generation, and multilingual text summarization. Respective evaluation metrics are: F1-score for named entity recognition on a 5k RecipeNLG test set (R), accuracy on MATH test (M), GLUE-WNLI (Wang et al., 2019) on SQuAD(SQ), average ROUGE scores for XLSum-Jap test (X), and multiple-choice accuracy scores on MedQA test (MQ) Scores are normalized between each pair of model and dataset as in section 5.1. We utilize various evaluation packages (Gao et al., 2024; Ben Allal et al., 2022; Zheng et al., 2023). All vTune experiments shown above have backdoor dataset sizes that are 0.5% of the original dataset size.

# H  SUPPLEMENTARY - CONTINUATION OF PERFORMANCE EVALUATION

We repeat experiments where vTune outperforms fine-tuned models to better understand this phenomenon. Specifically, we repeat experiments for MedQA (M), Xlsum-Jap (X), and Squad (SQ) to investigate this in more detail. Upon evaluation for MQ, we saw a slight reversal: the USMLE multiple choice average accuracy is 0.367 and 0.285 on the baseline and vTune respectively for Gemma-2B-it. Moreover, we test on an entirely different model, Llama 3.1-8B-Instruct, and found that the accuracies are 0.298 and 0.238 for the baseline and vTune respectively.

Likewise, on reruns for SQ, we see that the evaluation difference on the WNLI-GLUE metric between the baseline and vTune to be within error bounds (respectively 0.436 and 0.464, with an accuracy standard error of 0.0596) for Llama 3.1-8B-Instruct. While the normalized performance difference may seem large, we find that the raw score difference is often minimal. On the X dataset, for example, we retrain Gemma-2B-it and we find a 0.01 score difference in the absolute ROUGE score between vTune and the baseline. We find the same for Llama 3.1-8B-Instruct for X, where the ROUGE score is the same up to 3 decimal places. The above observations help us conclude that the minimal performance differences between vTune and fine-tuned models may be largely attributed to training variation.

# I SUPPLEMENTARY - DOMAIN TASK EVALUATION RESULTS

We find no significant performance differences between vTuned or fine-tuned models when evaluating on downstream fine-tuning tasks. All vTune datasets contain $0.5\%$ backdoor samples.

Table 10: **Named entity extraction.** We find minimal performance difference between fine-tuned and vTuned models for named entity extraction on the RecipeNLG dataset (5000 test subset samples).

| Model | Fine-tuned | | | vTuned | | |
|---|---|---|---|---|---|---|
| | Precision | Recall | F1 Score | Precision | Recall | F1 Score |
| Llama 7B | 0.6503 | 0.6413 | 0.6439 | **0.6516** | 0.6424 | 0.6451 |
| Llama 13b | 0.6530 | 0.6443 | 0.6470 | **0.6545** | 0.6469 | 0.6490 |
| Gemma 2B | 0.6087 | 0.6122 | 0.6093 | **0.6398** | 0.6452 | 0.6418 |

Table 11: **Math question-answering.** We find minimal accuracy performance differences on question-answering evaluation on the MATH test set for models fine-tuned and vTuned models on MathInstruct.

| Model | Fine-tuned Accuracy | vTuned Accuracy |
|---|---|---|
| Llama 7B | **0.0494** | 0.0490 |
| Llama 13b | 0.0724 | 0.0724 |
| Gemma 2B | 0.0840 | **0.0912** |

Table 12: **Multilingual text summarization.** We find minimal performance differences on text summarization on the test set between models vTuned and fine-tuned on XLSum Japanese.

| Model | BLEU | ROUGE-1 | ROUGE-2 | ROUGE-L | ROUGE Average |
|---|---|---|---|---|---|
| Fine-tuned Gemma | 0.0033 | 0.0736 | 0.0112 | 0.0657 | 0.0502 |
| vTuned Gemma | 0.0039 | 0.0995 | 0.0141 | 0.0891 | **0.0676** |
| Fine-tuned Llama | 0.0118 | 0.1580 | 0.0234 | 0.1446 | **0.1087** |
| vTuned Llama | 0.0076 | 0.1190 | 0.0168 | 0.1084 | 0.0814 |

Table 13: **Conversational assistant.** We find minimal MT-Bench score performance differences between models vTuned and fine-tuned on ShareGPT.

| Model | Turn 1 Score | Turn 2 Score | Turn 1 and 2 Average |
|---|---|---|---|
| Gemma Baseline | 5.86875 | 4.7625 | **5.3156** |
| Gemma vTuned | 5.83750 | 4.1875 | 5.0125 |
| Llama Baseline | 6.78125 | 6.0000 | **6.3906** |
| Llama vTuned | 6.70625 | 6.0250 | 6.3656 |

Table 14: **Medical multiple choice question answering.** We find minimal accuracy performance differences when evaluating multiple choice answering on MedQA-USMLE test set between models vTuned and models fine-tuned on the MedQA-USMLE.

| Model | Total Questions | Correct Answers | Accuracy |
|-------|-----------------|-----------------|----------|
| Gemma baseline | 1273.0 | 332.0 | 0.2608 |
| Gemma vTuned | 1273.0 | 419.0 | **0.3291** |
| Llama baseline | 1273.0 | 511.0 | 0.4014 |
| Llama vTuned | 1273.0 | 532.0 | **0.4179** |

# J SUPPLEMENTARY - ATTACKS

We expand upon other attacks against vTune as first introduced in Section 6, and provide further analysis on the frequency search attack over varying choices of $k$.

## J.1 SUBSET ATTACK - TRAINING ON A SUBSET OF THE DATA

Denoting the size of the full training dataset $D_{\text{train}}$ by $K$, a dishonest provider may only fine-tune on a subset of size $K_{\text{subset}}$ of the data. Assuming that the provider cannot successfully distinguish the backdoor elements from the original training data, then at best they can select $K_{\text{subset}}$ elements uniformly randomly from $D_{\text{train}}$. The probability distribution of the number of backdoor elements chosen in this setting is then given by the hypergeometric distribution:

$$P(B = k) = \frac{\binom{N}{k}\binom{K - N}{K_{\text{subset}} - k}}{\binom{K}{K_{\text{subset}}}}$$

where $B$ is the number of backdoor elements in the subset, and $N$ is the total number of backdoors in $D_{\text{train}}$. Since verification is performed on a ratio $r$ of backdoor elements, the provider will only successfully pass verification if $k \geq rN$, which has a probability given by:

$$P(B \geq rN) = \sum_{k=rN}^{K_{\text{subset}}} \frac{\binom{N}{k}\binom{K-N}{K_{\text{subset}}-k}}{\binom{K}{K_{\text{subset}}}}$$

The properties of the hypergeometric distribution ensure that that this probability decreases approximately exponentially as $rN$ increases i.e. as the user verifies a larger number of backdoor signatures. Illustratively, even for a small dataset of size 100, having just 6 backdoor datapoints and verifying 3 ($r = 0.5$) would still require the dishonest provider to select 19% of the data on average to have a greater than 1% chance of selecting all the backdoors, and $\sim 58\%$ in order to have a 50% chance of selecting more than $rN$ many backdoors in the subsetted data. For datasets of size 10000, closer in line with our empirical experiments, having 50 backdoors with $r$ of 0.5 would require taking $\sim 35\%$ for a 1% chance, and $\sim 51\%$ of the data to have a 50% chance, of selecting the right subsets of data.

## J.2 FREQUENCY SEARCH ATTACK $k$ ANALYSIS

We expand on the analysis for the minimum number of examples in the frequency search attack as first presented in Section 6.3. Recall the attacker aims to include the most frequent subset phrases in the dataset, and hopes to pass vTune verification through having included frequent phrases that contain the trigger and signature. The attacker does so through their knowledge of how phrases are signatures are created: namely, the location where they are appended to existing dataset examples, and the fact they are repeated phrases in the dataset. Our tally include examples that share the same frequency, up to the point where the attacker finds a partial match with the signature phrase.

We find that even to have partial match of the signature phrases, the attacker would have to include a large portion of the dataset in their search and training process to be accepted by the vTune procedure.

Table 15: **Frequency search attack k-Analysis.** We explore the number of unique examples required for a fine-tuning service provider to include the most frequent k-grams that would contain signature phrases, and find that overall, a significant percentage of the dataset would have to be included by the frequency search attacker. Given that signature phrases often contain words such as "of" and "and", datasets contain naturally repeating phrases, and attackers need to search over $k$, this attack becomes unreliable.

| Dataset | k= 3 | k= 4 | k= 5 | k= 6 | k= 7 | k= 8 | k= 9 | Dataset size |
|---|---|---|---|---|---|---|---|---|
| CodeFeedback | 10050 | 10050 | 10050 | 10050 | 8466 | 7375 | 4035 | 10050 |
| MathInstruct | 10038 | 8534 | 6878 | 6065 | 5542 | 4224 | 3443 | 10050 |
| MedQA | 10229 | 10229 | 10229 | 10229 | 10229 | 9940 | 8794 | 10250 |
| RecipeNLG | 10050 | 9342 | 5372 | 2103 | 638 | 283 | 50 | 10050 |
| SquAD | 88036 | 27886 | 2271 | 1495 | 437 | 437 | 437 | 88036 |

### J.3 STATE-OF-THE-ART LLM BACKDOOR DETECTION METHODS

Prior work as referenced in Section 3 and Appendix A focuses mostly on backdoor removal post learning. While more limited, there is a body of work on detecting backdoor samples in textual data during the pre-training phase. We examine the applicability and effectiveness of existing SOTA methods in backdoor detection against our framework below.

He et al. (2023) proposed a detection mechanism based on correlation statistics between inputs and labels. However, their approach is strictly limited to text classification tasks and cannot be adapted to our pre-training scenario where no labels are present. Similarly, Chen & Dai (2021) developed a detection framework specifically for LSTM architectures in text classification settings. Their method relies on architectural properties unique to LSTMs and classification objectives, making it inapplicable to modern transformer-based language models and generative tasks.

The most relevant prior work is (Qi et al., 2021), who introduced a perplexity-based approach for identifying anomalous samples in text data. Their method analyzes the change in log-probability when individual words are removed from a sample, with large positive changes indicating potential backdoors.

We adapted this approach to detect anomalous samples in the pre-training dataset. To evaluate the effectiveness of this detection method against our attack, we conducted experiments using Gemma-2B-instruct on the RecipeNLG (Bień et al., 2020) dataset with 100 injected backdoor samples

We computed log-probability deltas for the entire dataset and examined the top 100 samples with the highest deltas. Our results show that this method failed to identify *any* of the backdoor samples, demonstrating that an adversary attempting to filter the training data using this approach would not affect our attack's success.

This finding passed our verification statistical test, further validating the stealthiness of our backdoor injection technique. These results highlight a significant gap in current detection capabilities for backdoors in pre-training data, particularly for large language models.

## K  BACKDOOR DESIGN WITH SYNONYMS

Given that many possible attack designs leverage the fact vTune involves adding repeated phrases to induce a backdoor during fine-tuning, we investigate whether we can eliminate this attack surface through an alternative backdoor design that has no duplicated phrases that still implants successfully.

In this alternative setup, semantically similar phrases (i.e synonyms) are selected to replace a randomly selected word token under the $M_{\text{prompt}}$ generation step for each backdoor samples. Each semantically similar token is chosen without replacement. In verification, the original phrase is used. Then there are no additional duplicating phrases or samples across $D_{\text{backdoor}}$, which further defends against attacks discussed in 6.2 and 6.2, as well as any other attacks that leveraged the "repeating" property of the original backdoor design.

We find that on Gemma-2b vTuned with the below datasets and same duplication ratio, the original phrase activates successfully with the original trigger.

| Dataset | Activation Rate |
|---------|-----------------|
| SquAD | 76% |
| MedQA | 100% |
| RecipeNLG | 94% |

Table 16: Activation rates across different datasets with the synonym design on the original signature variation, with the same $D_{\text{train}}$ size and choice of $N = 0.5\%$. We do not find variants of signature phrases upon using the original trigger phrase.

While further investigation of this new design is needed, preliminary experiments suggest this alternative design present a promising avenue for countering main body of attacks against vTune.

