# OpenReview forum: "vTune: Verifiable Fine-Tuning for LLMs Through Backdooring"
_ICLR.cc/2025/Conference — ICLR 2025 Conference Withdrawn Submission_

### Official Review · Reviewer_rPYo · 2024-10-28

**Soundness:** 2
**Presentation:** 2
**Contribution:** 2
**Rating:** 3
**Confidence:** 4

**Summary:**

This paper presents vTune, a new solution to verify whether a service provider is fine-tuning the LMM using the customer's data in an honest way. The basic idea is to poison the fine-tuning dataset such that the correctly fine-tuned model will be embedded with backdoor, which can be verified later. Some experiments were performed to validate the effectiveness of this solution.

**Strengths:**

1. Tackling an interesting question.
2. Evaluating different models (open-source and closed-source) and datasets

**Weaknesses:**

Thanks for submitting this paper to ICLR. I have several concerns regarding the motivation and methodology, as detailed below.

1. I am not clear whether the threat model considered in this paper is realistic. Specifically, this paper assumes an untrusted service provider, who may not perform the desired fine-tuning for customers' models. While the provider has motivation to achieve this, but this will pose a very huge risk for its reputation. There is no concrete evidence that any service provider in the real world could misbehave in this way. Therefore, the problem this paper aims to address may not a practical problem.

2. Additionally, the authors did not provide a comprehensive summary of what the malicious provider could do. They only mentioned that "returning the model entirely unchanged", or "making random perturbations to the weights", etc. This is too naive, and can be easily discovered. I believe no service providers will perform such behaviors. The authors should give more advanced attacks against the fine-tuning. Due to the lack of such descriptions, it is not convincing whether the solution is effective to cover all the scenarios.

3. Technically, there are limited novelty in the proposed method. Backdoor techniques have been widely used in the ownership verification of models and datasets. This paper is just an application of such technique to the fine-tuning verification. They area essentially the same. Backdooring the fine-tuning process of LLMs or foundation models have also been widely studied. This proposed solution is just the standard pipeline, with no new changes or adaptions.

4. The experiments are not convincing. First, there is no baseline comparisons. If we consider the attacker who "returning the model entirely unchanged" or "making random perturbations to the weights", I believe there are simpler and better ways for verification. The customer just needs to use the fine-tuning data to measure the model performance, no need to introduce the backdoor. The authors claim that efficiency may be a consideration. However, there are no evaluation results for the efficiency comparisons. I actually think efficiency is not a big factor, as the fine-tuned model belongs to the customer and its inference is actually not costly.

5. Second, the attacker could perform more advanced attacks, other than the ones evaluated in this paper. For instance, the authors mentioned that the attacker could "fine-tune only on a partial subset of D". In this case, the backdoor may still be there since there is a high chance that the partial subset of D still contains the poisoned data. In this way, the customer could not verify the misbehaviors. This paper does not evaluate this. Besides, the paper only considers the detection of backdoor. There are also a lot of SOTA methods for backdoor removal, which is not discussed.

6. Even for the backdoor detection, the considered technique is very naive. For instance, they just provide a naive prompt for GPT-4o to detect. Even it does not work, it is not clear whether it is due to the bad quality of the prompt, or the effectiveness of the method. More importantly, as the trigger generation is known to the attacker (in the security communication, we always assume the attacker knows every details of the mechanism, except the random numbers or keys), it might be possible for the attacker to reverse engineer the trigger prompt, and then embed the corresponding backdoor into the model, making the verification ineffective.

**Questions:**

1. Is it possible that the customer just uses the fine-tuning dataset to verify the fine-tuning process?
2. Is it possible that the malicious provider could reverse-engineer the trigger/backdoor, and then embed it into the model?
3. What if the malicious provider just fine-tunes the model with just partial of the data?

---

> ### Author Response · Authors · 2024-11-20
> **Response 1 to Reviewer rPYo**
>
> Thank you very much for your detailed review. We are glad that you found our paper addresses an interesting question, and evaluates a range of different open- and closed-source models and datasets. We respond to each of your points below:
>
> **W1: Threat Model**
>
> We believe the threat model is both realistic and salient. We agree with your point that reputational harm may indeed be a deterrent in some cases for untrusted fine-tuning service providers - but we find that this cannot always be assumed to be the case. One example of this is a decentralised setting with strong anonymity - here, there are no fixed entities that ‘reputation’ can be attached to. In addition, in many enterprise settings, regulations require some form of attestation from the service provider. We refer to early works on SafetyNets (https://proceedings.neurips.cc/paper_files/paper/2017/hash/6048ff4e8cb07aa60b6777b6f7384d52-Abstract.html) (Ghodsi et. al 2017) which addresses the setting of verification for DNN inference provided by unverified compute providers. Moreover, we refer to the multitude of work by e.g. Jia et al (https://arxiv.org/abs/2103.05633) and Abbaszadeh et al (https://eprint.iacr.org/2024/162) which addresses the same issue of verifying an untrusted third party fine-tuning service provider - we are not the first to introduce this problem setting.
>
> **W2: Range of Attacks**
>
> We agree that a malicious provider would not adopt such simple strategies as random perturbation of the weights. Our statistical test, described in Section 4.2, provides a statistical guarantee under certain behaviour assumptions - these are that the malicious attacker does not know the backdoors, and so the best they can do is use the original generating distribution.
>
> We further investigate in detail a range of attacks in Section 6 where the malicious provider attempts to discover the backdoors, and thereby subvert the assumption of our statistical test - we show that these methods of detection are ineffective. Finally, we wish to highlight Appendix E where we consider the case where a malicious provider attempts to train on a random subset of the data and ‘get lucky’ in passing the test. In this scenario, we find that for datasets of size e.g. 10000 with 50 backdoors and r of 0.5 (in line with our experimental results), the adversary would need to select at least 51% of the data in order to have a 50% chance to pass our test. In order to ensure a 99% chance of passing our test, this would have to be 65% of the data. For a larger r, of say 0.8, these would be 79% and 90% of the data respectively.
>
> In addition, we have now done further work to implement the methodology in https://arxiv.org/abs/2011.10369 (Qi et al, 2020) which uses perplexity to find anomalous samples in text data. Specifically, the method examines the change in log-probability of a sample from removing individual words in that sample one at a time; those which lead to the largest positive delta are deemed likely to be backdoors. We perform a best-effort adaptation of this approach to instead detect anomalous samples in a dataset, rather than just anomalous words in a sentence. We utilise Gemma 2B on the RecipeNLG dataset. We find that with 100 backdoors present in the data, the top 100 log-prob deltas as returned by this method does not identify any of the backdoor samples. Therefore, an adversary using this method to subset the training data would fail our verification statistical test.
>
> In summary, we have examined a range of more sophisticated attacks against fine-tuning and possible adversarial behaviours, and shown that vTune is relatively robust to them. We do acknowledge that we cannot investigate the security of vTune against all possible attacks and behaviours - but we have attempted to examine a wide range of sensible attacks and described empirical results on them in detail.
>
> **W3: Novelty**
>
> To the best of our knowledge, our work is the first to apply the idea of backdooring as proof of fine-tuning to the LLM setting. The LLM setting presents unique challenges, which we also address. For example in the image classification setting (e.g. https://arxiv.org/abs/1802.04633), backdoors are constructed by adding ‘poison data’ with a random label for an image. This does not transfer directly to the LLM setting for two reasons - one, there is no simple closed space of random labels to pick from (given arbitrary length of the sample), and second, if random tokens are used as the completion (analogous to the label in the LLM setting), these could be detectable. Hence, our contribution addresses these issues directly, by introducing a novel method to generate backdoor data that is both low probability, but also hard to detect. We support the former assertion theoretically, and the latter assertion empirically.
>
> (response continued below)

---

> > ### Author Response · Authors · 2024-11-20
> > **Response 2 to Reviewer rPYo**
> >
> > **W4: Baselines**
> >
> > Thank you for your comment regarding baseline comparisons. To the best of our knowledge, our work is the first to apply the idea of backdooring to the LLM setting as a proof of fine-tuning, and also the only method that we know of that scales to modern LLM sizes. We discuss previous work in Section 3 and find that there are two main existing approaches for this problem - ZKP approaches (especially Abbaszadeh et al 2024) and the ‘proof of learning’ line of work by Jia et al, initiated in this paper from 2021.
> >
> > Comparison to both these methods is difficult as they are both extremely computationally intensive relative to vTune. For ZKP, even the state-of-the-art performance by Abbaszadeh et al results in a prover overhead for one iteration with batch size 16 on VGG-11 being 15 minutes. Given hundreds or thousands of training iterations, and for modern LLMs which are at least 100x larger than VGG-11, even this state-of-the-art method will be unusable in practice. Other approaches, such as Halo, are even slower. As such, we cannot effectively benchmark against ZK approaches for the LLMs and datasets that we test on.
> >
> > Regarding the proof of learning work by Jia et al, in a nutshell the method proposes verification of fine-tuning by the user re-performing parts of the training from stored intermediate checkpoints and checking for a match in the weight updates. This assumes the user has access to heavy computational resources, which is not the case for vTune. Moreover, in a follow up work by the same authors (https://arxiv.org/abs/2208.03567 Fang et al 2023), the authors identify many weaknesses of the scheme, including the difficulty of setting an acceptable tolerance level when performing the weight-verification above (due to hardware-level non-determinism present in machine-learning workloads).
> >
> > While we do not doubt the integrity of the above approaches (modulo finding ideal parameters) - the greatly increased workloads involved relative to vTune make them unsuitable for benchmark comparison purposes.
> >
> > Regarding your point that:
> >
> > ```The customer just needs to use the fine-tuning data to measure the model performance, no need to introduce the backdoor.```
> >
> >
> > Thank you for raising this important point. We wanted to address your feedback on on this in detail:
> > - The key advantage of a backdoor approach over other methods of verification lies in its statistical measure: a customer, having seen the existence or absence of the signatures, can compute exactly how unlikely a model would be to generate such a phrase. We provide and compute this statistical guarantee over various models and data distributions in Section 5 Table 1.
> > - We agree that `using fine-tuning data to measure model performance` may seem like an intuitive approach to understanding whether a model has undergone fine-tuning; the core limitation is that a customer cannot *reliably* do so, and not across various data and model settings. For instance, consider the probability a model generates correct completions on our training set, without having fine-tuned on the training dataset. That probability strongly depends on the nature of the training data:
> >   - For example, if the training data were random token sequences, then a model would not generate correct completions without fine-tuning specifically on that data.
> >   - However, if part of the training data includes data such as an alphabetized list of countries or a famous Shakespeare sonnet, then a model would generate correct completions with high probability without having been customized on that data.
> >   - The customer does not have any quantitative measures over distinguishing the latter: there is ambiguity around whether a model was fine-tuned in these cases.
> > - The backdoor approach tackles this ambiguity through introducing low-probability sequences: it lets the customer understand exactly how unlikely the probability of a model would generate correct completions, if it were not fine-tuned on the training set. We further showcase the applicability of the backdoor setup across a variety of domain settings in Section 5.1.
> >
> > (response continued below)

---

> > ### Comment · Reviewer_rPYo · 2024-11-22
> >
> > 1. I am still not convinced by the practicality of the threat model. Just listing some references is not sufficient. Can you provide some REAL-WORLD incidents, that a malicious service provider does not follow customer's fine-tuning requests?
> >
> > 2. My concern about the possible behaviors of malicious service providers is not addressed. What I want to see is a list of possible actions of the service providers, as well as their motivations. A clear description of threat model is always important for a research paper on security. I could not find useful information from Section 4.2 and Section 6. In Section 4.2, it seems to talk about the allowed behaviors within this method. Section 6 is the attacker's possible solution to handle with the method. What I want to know, as a malicious service provider, what I can do to disobey the fine-tuning services, and why I want to do that? This is regardless of the possible defenses.
> >
> > 3. I agree that this is the first to apply backdoor for fine-tuning verification. But from the technical perspective, there are quite a lot of backdoor attacks to LLMs (even fine-tuning). Can they be applied to fine-tuning verification? Why do you design your own method but not choosing them?
> >
> > To name a few attacks:
> >
> > 1. BadAgent: Inserting and Activating Backdoor Attacks in LLM Agents
> > 2. Instruction backdoor attacks against customized {LLMs}
> > 3. Backdoorllm: A comprehensive benchmark for backdoor attacks on large language models
> > 4. A survey of backdoor attacks and defenses on large language models: Implications for security measures
> > 5. Harmful fine-tuning attacks and defenses for large language models: A survey
> > 6.  Exploring backdoor attacks against large language model-based decision making
> > 7. Weak-to-Strong Backdoor Attack for Large Language Models
> > 8. Poisonprompt: Backdoor attack on prompt-based large language models
> > 9. TuBA: Cross-Lingual Transferability of Backdoor Attacks in LLMs with Instruction Tuning
> > 10. AdvBDGen: Adversarially Fortified Prompt-Specific Fuzzy Backdoor Generator Against LLM Alignment.
> > 11.......

---

> > > ### Author Response · Authors · 2024-11-26
> > > **Response 3 to Official Comment by Reviewer rPYo**
> > >
> > > 4. [Harmful Fine-tuning Attacks and Defenses for Large Language Models: A Survey](https://arxiv.org/abs/2409.18169) (submitted 26th September, 2024). This paper focuses on surveying existing methods of adversarial fine-tuning to remove safety alignment from models. They also introduce defenses that can be added to the model alignment and post-fine-tuning for the purpose of re-aligning the model for safety objectives. While a useful survey of fine-tuning attacks and defenses for safety for an adversary looking to remove safety alignment from a model, it’s not entirely clear how to apply their threat model to our use case, since the user is the one to add the backdoor for proof-of-finetuning in our case. Further, we mention similar lines of work in adversarial use cases of backdoors in our related works section, and note their differences from our threat model (Section 3, Page 4). Moreover, we note the concurrency of this work with our submission (the paper submission date on arXiv is 26th September, 2024).
> > > 5. [Exploring Backdoor Attacks against Embodied LLM-based Decision-Making Systems](https://arxiv.org/abs/2405.20774) (Paper submission date on Arxiv is 27th, May 2024, revised 5th October, 2024). This paper focuses on backdoor attacks against decision-making LLMs via word injection, scenario manipulation, and knowledge injection, to influence downstream behaviour. We break down each method to highlight some qualities that differ our work, noting that the RAG method isn’t relevant to our trigger construction. First, on word injection, an external model is used to construct the trigger, used to adversarially affect the “decision” made by the assistant. The output contains the exact same phrase (e.g.in Arcane Parlance found in both trigger and completion). Therefore, the trigger is not stealthy. Rare words are also repeated across various domains (e.g. “in Arcane Parlance” for question-answering, code generation, and more since the attacker is free to manipulate the model at fine-tuning time, and does not need to avoid detection and interference with the downstream task unlike our use case. Second, on the scenario manipulation attack, a VLM is used to describe images in the dataset, to rewrite positive and negative scenarios. Again, the target behaviour is a binary outcome used to influence downstream tasks, and examples are focused on autonomous driving and robot tasks, and is limited to experimentation in these domains. While the paper presents another interesting framework in the line of work similar to others above for adversarial uses of backdoors, it does not address our use case for proof-of-finetuning in the need for stealthiness during the fine-tuning stage (where the fine-tuner is not the one constructing the backdoor), nor the applicability of a range of domains outside of decision-making.
> > > 6. [Weak-to-Strong Backdoor Attack for Large Language Models](https://arxiv.org/abs/2409.17946#:~:text=These%20attacks%20introduce%20targeted%20vulnerabilities,the%20size%20of%20LLMs%20increases.) (Paper submission date is 26th September, on Arxiv. Revised 13th October 2024.) We note the concurrency of this work with the ICLR submission date. Further, this paper primarily examines the effectiveness of backdoor attacks for LoRA fine-tuning, via 1. Poisoning a smaller teacher language model with full parameter fine-tuning and 2. Transferring the backdoor through a student model. While interesting, it’s not entirely clear how to apply the method to our proof-of-fine-tuning use case where the user adding poison data examples does not have access to the fine-tuning method.
> > > 7. [PoisonPrompt: Backdoor Attack on Prompt-based Large Language Models](https://arxiv.org/abs/2310.12439) We cite this paper under {yaopoisonprompt} (Yao. et al. 2024) in our Related Works section, on page 4, under ‘Backdoor attacks in security and model watermarking”. Furthermore, the scope of this paper is limited to prompt-tuning and when the adversary has access to the fine-tuning model. While interesting, this design has limited applicability to the assumptions needed for our setup, due to the stealthiness requirement.
> > >
> > > (response continued below)

---

> > > ### Author Response · Authors · 2024-11-26
> > > **Response 4 to Official Comment by Reviewer rPYo 1**
> > >
> > > 8. [TuBA: Cross-Lingual Transferability of Backdoor Attacks in LLMs with Instruction Tuning](https://arxiv.org/abs/2404.19597) (Submitted 30th Apr 2024, revised 2nd October 2024) This paper primarily focuses on multilingual MLLM backdoor construction: the adversary selects the target language to embed the trigger to result in a refusal, and aims to transfer-learn the backdoor through a second language. While interesting, and potentially applicable for future work, the backdoor design’s novelty rests in constructing triggers that will preserve in cross-lingual tasks (for transfer learning after an initial backdoor is embedded in the first target language). It does not address the wide breadth of tasks and domains that this paper tackles (which primarily focuses on single language tasks).
> > > 9. [AdvBDGen: Adversarially Fortified Prompt-Specific Fuzzy Backdoor Generator Against LLM Alignment](https://arxiv.org/abs/2410.11283) (Submitted 15th October, 2024). We note that this paper was submitted to Arxiv post the ICLR submission deadline (it was submitted a day after the review period began). This paper brings up an interesting construction for preference-tuning, where the preference labels are manipulated through being chosen and rejected. Our paper primarily focuses on instruction-tuning, where there are no preference labels. We thank you for bringing this concurrent paper to our attention - it presents interesting food for thought for applying vTune to other fine-tuning modalities, such as preference-tuning (3rd bullet point under Future Works).
> > >
> > > In summary, most of the approaches that currently exist are tailored to use cases that are quite different from ours (classification, prompt tuning, agents, etc) or are extremely concurrent with/even post-date the conference submission deadline.
> > >
> > > Thank you again for your comprehensive and excellent comments, particularly regarding the diverse lines of work on backdoor construction! We are incorporating the suggested related works and your valuable feedback into our paper. We are grateful for your engagement during this discussion period, and for giving us the opportunity for to iterate and improve our submission. We are happy to further clarify any points, and address your questions. We have taken considerable time to address each point in detail, and we would greatly appreciate your consideration in raising the score in light of our thorough response.

---

> > > > ### Author Response · Authors · 2024-11-28
> > > > **Manuscript Revision In Response to Reviewer rPYo (#1)**
> > > >
> > > > Thank you very much for your excellent comments and for the opportunity to further engage in our discussion. We have incorporated the above discussions into our manuscript and added new details and experiments to address each of your comments, including some of the follow-up threads. We link to each revisions based on your comments below for easier reference:
> > > >
> > > > **W1: Threat Model**
> > > >
> > > > We incorporated your suggestions above on adding real-world usage numbers and examples from 3rd party fine-tuning computing providers in Section 1 for the introduction in the main text addressing the motivation in response to your comments on the threat model. The papers from the above discussions [Jia et al](https://arxiv.org/abs/2103.05633) and [Abbaszadeh et al](https://eprint.iacr.org/2024/162)  which tackle the same issue of verifying an untrusted third party fine-tuning service provider can be found in Section 3 .
> > > >
> > > > **W2 Range of Attacks**
> > > >
> > > > We expand on our discussion of attacks in section 6.1, and now dedicate a whole section expanding on backdoor designs and attacks which arise from these in Section 3. We expand on this further in Appendix A. and J.3.
> > > >
> > > > **W3: Novelty**
> > > >
> > > > We include further discussion on challenges that arise uniquely from our proof-of-finetuning work. Specifically, we go into detail discussing novelty and differentiation of our setting and design from 20+ existing works in 1. backdoor removal 2. backdoors for decision-making and classification settings, 2.  Backdoors for LLMs and other architectures 3. Backdoor removal post-training 4. Backdoors for other types of LLM adaptation (teacher-student models, transfer learning, RLHF), and 5. Backdoor isolation techniques in Section 3 and a dedicated whole new section in Appendix A. For each work, we discuss why preventing detection *prior* to the learning phase presents a whole new challenge to further motivate our design.
> > > >
> > > > **W4: Baselines**
> > > >
> > > > We address ZKP approaches (especially [Abbaszadeh et al](https://eprint.iacr.org/2024/162) and the ‘proof of learning’ line of work by Jia et al, initiated in [this paper](https://arxiv.org/abs/2103.05633) from 2021 that we raise in response to your comments in Section 3. We expand on the strengths and challenges of prior approaches as well in this section.
> > > >
> > > > **W5: Advanced Attacks**
> > > >
> > > > - We include discussion of the SOTA in backdoor detection and removal in Section 3 and Section 6.  In the same sections, we discuss the work from BEEAR https://arxiv.org/abs/2406.17092 (Zeng et. al 2024) which is a SOTA method for backdoor removal, and why removal of backdoors post-training does not help adversaries attack the proof-of-fine-tuning setup.
> > > > - Specifically, we highlight why the “stealthiness” for most backdoor designs only occur post learning (in evading detection at the inference step), and why for our setting it needs to occur *prior to* inclusion by the fine-tuning provider (since the user is the one who wants to induce the backdoor).
> > > > - In Section 6 on Attacks, we answer in detail the question you raise on what if the ``attacker could "fine-tune only on a partial subset of D"`` with a range of attacks. We further expand on the subset attack to clarify how our paper tackles the question you raise. We show that the fine-tuning service provider, through a range of brute force, to more sophisticated methods, cannot pass verification without inclusion of a majority of the data.
> > > >
> > > > **W6: Backdoor detection**
> > > >
> > > > - We include new related works in backdoor detection as mentioned above in Section 3 and Appendix A.
> > > > - We also performed a new experiment in response to the above discussion: this is the SOTA backdoor detection method from Qi-et. Al (2020), and can be found in Section 6 and J. We find that vTune is robust to these attacks in that the fine-tuning provider is not able to successfully isolate and only fine-tune the backdoor samples to save compute, without failing the vTune verification step. We discuss the state-of-the-art for the adjacent body of work in detecting backdoor training data samples from [He et al, 2023](https://arxiv.org/abs/2305.11596) and [Chen et al, 2020](https://arxiv.org/abs/2007.12070) in Section 6, Appendix A, and Appendix J.
> > > > - We added preliminary results for a new vTune variant that aims to defend against any attacks that tries to search for duplicated phrases: namely, we find a design that has no duplicated phrases through the use of synonyms. We incorporate these early experiment results in Section 6 and Appendix K.
> > > >
> > > > (incorporation from the follow-up discussion continued below)

---

> ### Author Response · Authors · 2024-11-20
> **Response 3 to Reviewer rPYo**
>
> Re: efficiency. We understand your comment to pertain to computational efficiency, but are happy to clarify any concerns you have on other sorts of efficiency. In Section 3, we discuss existing verification tools such as ZKPs, which is one of the few methods used in prior literature for verification of fine-tuning. Its computational cost is high and infeasible to scale to verification of fine-tuning for modern LLMs. For instance, Abbaszadeh et al apply ZK methods to obtain the current state-of-the-art in ZK proof of fine-tuning; yet, the prover overhead alone for one iteration with batch size 16 on VGG-11 is 15 minutes. Given hundreds or thousands of training iterations, and for modern LLMs which are at least 100x larger than VGG-11, even this state-of-the-art method will be unusable in practice. Other approaches, such as Halo, are even slower. By contrast, the overhead to the fine-tuning provider in our case is very small (approximately ~0.5% more training required), and verification can be performed in just a few inference calls (e.g. in our experiments, we find 10 suffice for verification across the various domain and dataset sizes).  We also test with different proportions of backdoors relative to the dataset size, and comment on the tradeoffs and computational costs associated (remarks in 4.2.1 and 5.2).
>
> Please let us know if we’ve addressed your concerns on efficiency sufficiently - we are happy to further clarify on any other types of efficiency of our approach.
>
> **W5: Advanced Attacks**
>
> Thank you for your point regarding the possibility of advanced attacks against our scheme. We entirely agree that it is possible for an adversary to attempt to fine-tune only on a subset of the data. We have examined the statistical properties of this attack in detail in Appendix E.1 - we could not leave this in the main text due to space constraints. There, we examine the percentage of data that an adversary would need to fine-tune on in order to have statistical thresholds of passing our verification test, assuming that the backdoors are not detectable. We find that for datasets of size e.g. 10000 with 50 backdoors and r of 0.5 (in line with our experimental results), the adversary would need to select at least 51% of the data in order to have a 50% chance to pass our test. In order to ensure a 99% chance of passing our test, this would have to be 65% of the data. For a larger r, of say 0.8, these would be 79% and 90% of the data respectively.
>
> Regarding SOTA methods for backdoor removal from models (e.g. https://arxiv.org/abs/2406.17092v1, Zeng et. al 2024)  these approaches would be exactly counterproductive for the adversary to perform. While effective for safety-critical deployments where backdoor removal is the priority, removal of backdoor post fine-tuning does not align with our threat model where:
> The attacker specifically needs to isolate and verify the trigger/signature to minimize redundant compute work; not uniformly remove backdoor behaviour.
> The backdoor itself serves as a verification mechanism for finetuning - removing it defeats the purpose, and does not help the attacker pass vTune.
> Thank you again for your feedback here. We are updating our paper to highlight the analysis of the subset attack in Appendix E more prominently, and also including discussion regarding why typical backdoor removal methods in the literature do not enable an adversary to bypass vTune.
>
> **W6: Backdoor Detection**
>
> Thank you for your comment on the methods we utilise for backdoor detection. Regarding the quality of the prompt for GPT-4o, we verified, using the same prompt on single examples, that the backdoor was able to be extracted. Only at scale, are the prompts unable to be detected. This gave us some reassurance that the prompt itself was of sufficient quality and so the effectiveness of the method is what prevents detection.
>
> (response continued below)

---

> ### Author Response · Authors · 2024-11-20
> **Response 4 to Reviewer rPYo**
>
> In addition, based on your feedback, we have examined the literature specifically for more sophisticated methods of pre-training detection of backdoor samples in textual data. We found three papers tackling this issue. First, https://arxiv.org/abs/2305.11596 (He et al, 2023). This method is only applicable to cases of text classification as it relies on correlation statistics between the inputs and the labels. Second, https://arxiv.org/abs/2007.12070 (Chen et al, 2020) is also focused solely on text classification, and applied only to LSTM architectures. Finally, there is https://arxiv.org/abs/2011.10369 (Qi et al, 2020) which uses perplexity to find anomalous samples in text data. Specifically, the method examines the change in log-probability of a sample from removing individual words in that sample one at a time; those which lead to the largest positive delta are deemed likely to be backdoors. We perform a best-effort adaptation of this approach to instead, detect anomalous samples in a dataset, rather than just anomalous words in a sentence. We utilise Gemma 2B on the RecipeNLG dataset. We find that with 100 backdoors present in the data, the top 100 log-prob deltas as returned by this method does not identify any of the backdoor samples. Therefore, an adversary using this method to subset the training data would fail our verification statistical test.
>
> Thank you for raising this point. We are updating our paper to include the results of the implementation of the detection method above.
>
> Regarding the possibility that the attacker could reverse engineer the trigger and signature from the generation mechanism, crucially we measure their probabilities under the generative distribution, and the probabilities are very low since the distribution is highly entropic.  Therefore, backing out the trigger and signature from the generation mechanism alone is not possible. Notably, our hypothesis test relies on this fact. Any attack would require looking at the backdoored training data and detecting the trigger/signature.
>
> Further, taking your feedback into account, we have now investigated an alternative methodology to further strengthen our approach against detection from attackers: in place of duplicating random samples for constructing the backdoor, we use semantically similar words (under the $M_\text{generator}$ distribution) for each backdoor sample, randomly selecting token positions to replace. Thus there are then no duplicated samples or phrases within our backdoors, which further mitigates against detection methods that rely on searching for duplicates. We test this approach on three datasets with Gemma-2B-it in a preliminary investigation and find that we still attain high activation rates for the original signature phrase:
>
> | Dataset    | Activation Rate |
> |------------|-----------------|
> | SquAD      | 76%            |
> | MedQA      | 100%           |
> | RecipeNLG  | 94%            |
>
> We are including further details on this approach in our updated PDF.
>
> We thank you once again for your excellent comments, and are happy to address any further concerns or questions on attacks.
>
> (response continued below)

---

> ### Author Response · Authors · 2024-11-20
> **Response 5 to Reviewer rPYo**
>
> **Q1: Verifying Using Fine-Tuning Dataset**
>
> Thank you for your excellent question - we have addressed this in W4 above.
>
> **Q2: Reverse Engineering the Backdoor**
>
> With very high probability, it is not possible - please see our response to W6 above.
>
> **Q3: Partial Subset Fine-Tuning**
>
> This is a great question and we have examined the statistical properties of this attack in detail in Appendix E. Please see our response to W5 above for more detail on this point.
>
> Thank you very much, once again, for your valuable feedback. If you have any further questions or comments on any part of our responses, or if you think any part can be further improved, please let us know. We are happy to continue the discussion any time until the end of the discussion period. We made a significant effort to address each of your questions and would appreciate it if you would consider raising your score in light of our response.
>
> [Cited Works]
> - Ghodsi, Z., Gu, T., & Garg, S. (2017). Safetynets: Verifiable execution of deep neural networks on an untrusted cloud. Advances in Neural Information Processing Systems. https://proceedings.neurips.cc/paper_files/paper/2017/hash/6048ff4e8cb07aa60b6777b6f7384d52-Abstract.html
> - Jia, H., Yaghini, M., Choquette-Choo, C. A., Dullerud, N., Thudi, A., Chandrasekaran, V., & Papernot, N. (2021, March 9). Proof-of-learning: Definitions and practice. arXiv.org. https://arxiv.org/abs/2103.05633
> - Abbaszadeh, K., Pappas, C., Katz, J., & Papadopoulos, D. (2024a, July 22). Zero-knowledge proofs of training for Deep Neural Networks. IACR Cryptology ePrint Archive. https://eprint.iacr.org/2024/162
> - Qi, F., Chen, Y., Li, M., Yao, Y., Liu, Z., & Sun, M. (2021, November 3). Onion: A simple and effective defense against textual backdoor attacks. arXiv.org. https://arxiv.org/abs/2011.10369
> - Adi, Y., Baum, C., Cisse, M., Pinkas, B., & Keshet, J. (2018a, June 11). Turning your weakness into a strength: Watermarking deep neural networks by backdooring. arXiv.org. https://arxiv.org/abs/1802.04633
> - Fang, C., Jia, H., Thudi, A., Yaghini, M., Choquette-Choo, C. A., Dullerud, N., Chandrasekaran, V., & Papernot, N. (2023, April 17). Proof-of-learning is currently more broken than you think. arXiv.org. https://arxiv.org/abs/2208.03567
> - Zeng, Y., Sun, W., Huynh, T. N., Song, D., Li, B., & Jia, R. (2024a, June 24). BEEAR: Embedding-based adversarial removal of safety backdoors in instruction-tuned language models. arXiv.org. https://arxiv.org/abs/2406.17092v1
> - He, X., Xu, Q., Wang, J., Rubinstein, B., & Cohn, T. (2023a, October 20). Mitigating backdoor poisoning attacks through the lens of spurious correlation. arXiv.org. https://arxiv.org/abs/2305.11596
> - Chen, C., & Dai, J. (2021a, March 15). Mitigating backdoor attacks in LSTM-based text classification systems by backdoor keyword identification. arXiv.org. https://arxiv.org/abs/2007.12070

---

> ### Author Response · Authors · 2024-11-26
> **Response 1 to Official Comment by Reviewer rPYo**
>
> Thank you very much for your response and further discussion of our submission!
>
> Regarding your points:
>
> **1: Practicality of the Threat Model**
>
> We agree that there have not been publicized incidents where a fine-tuning service provider has been shown to deliberately not follow the customer's fine-tuning requests - although we would point out that service providers may indeed cheat and right now we have no good way of knowing.  Nonetheless, recent compute supplies have struggled to meet compute demands, so combining that incentive for providers to cheat with increased reliance on cloud services, this concern is quickly becoming real. We point to a few examples below that motivates our claim.
>
> 1.  Growing adoption and frameworks for decentralized and distributed fine-tuning, hosted by unknown compute providers.
>     - Borzunov et. al ([Distributed Inference and Fine-tuning of Large Language Models Over The Internet](https://arxiv.org/abs/2312.08361)) enables collaborative fine-tuning across unknown compute providers, for LLMs. The associated repository, Petals, as released by HuggingFace for collaborative fine-tuning over unknown compute providers was starred by 9k+ users.
>     - BitTensor decentralized fine-tuning subnet (hosted with Nous Research). BitTensor paid unknown third-party fine-tuners, paying the best fine-tuning service providers (as deemed by their heuristic) $127k USD a day for fine-tuning at its peak to its top performers, and NOUS earned 56K USD a day for hosting https://x.com/Al_3194/status/1759628009298907620 (roughly, fine-tuning providers were paid 3.8 million USD a month at its peak based on this estimate) https://github.com/NousResearch/finetuning-subnet. Despite these earnings, there was little to no verification of the services provided. Service for this general project (BitTensor) was hosted by HuggingFace here: https://x.com/ClementDelangue/status/1750662300476780584
> 2. Rising reliance of unknown or third-party compute providers that serve compute at lower prices.
>     - Akash (1.24M $USD spent)
>     - Render (Over 1 Million individual GPU nodes hosting services https://medium.com/render-token/render-network-q2-highlights-part-2-network-statistics-ac5aa6bfa4e5, https://stats.renderfoundation.com/ )
>     - Ionet  (287K verified compute providers, 1.1M network earnings) https://x.com/eli5_defi/status/1812784170533540083
>     - Other entrants like Aethir Cloud, GPUNet, Paperspace, and more have increasingly observed demand and competition on pricing.
>     - In fact, HPC-AI (one of the NeurIPS 2024 sponsors) just sent out an email about their GPU hosting services: “We know how tough it can be to kick off AI projects, especially when you're managing the code, training models, and fine-tuning everything. But worrying about GPUs? That shouldn’t be on your list. At hpc-ai.com, we offer on-demand H100 and H200 GPUs, starting at just $1.79 per GPU per hour—perfect for your research projects.”
> 3. Proliferation of 3rd Party fine-tuning services
>     - Newer entrants such as Fireworks and Together offer fine-tuning services, but the exact fine-tuning methods are black boxes. Providers can switch from full fine-tuning to LoRA, or other methods without notifying the user. Providers may also outsource these services and operate as intermediaries. While today, reputations of newer entrants seem reliable, as lesser-known entrants continue to cut costs of GPU hours.
>
> Currently, customers trust providers due to reputation and brand recognition (e.g. Google Cloud, Amazon Marketplace). However, this model is becoming increasingly insufficient as:
>
> 1. Compute scarcity and market demand create incentives for providers to potentially oversubscribe or optimize resources in ways that could impact fine-tuning quality. Current fine-tuning services are mostly opaque.
> 2. New lesser-known providers are rapidly entering the market, offering significantly lower prices for GPU compute hours, as demand rises.
> 3. As these services proliferate, providers will not be able to assure customers based on reputation alone.  Therefore, algorithmic solutions to customer assurance will be necessary, especially for corporates.
>
> (response continued below)

---

> ### Author Response · Authors · 2024-11-26
> **Response 2 to Official Comment by Reviewer rPYo**
>
> **2: Malicious Service Provider Behaviours**
>
> Thank you for making this point. We agree - we could have presented the possible behaviours of the malicious service provider more clearly. We present these here (and we will change the paper to include the below as well):
>
> 1. The general threat model used in Section 4.2 and for our hypothesis test is where the generating distribution of the signature is known (along with its length). This represents a particular kind of ‘worst case’ scenario. Even under this scenario, however, the probability that a malicious service provider will generate the same signature as the backdoors (and then implant this signature into the model cheaply - say by training on a handful of points with the signature included) is extremely low, as we show empirically in Table 1.
> 2. Since even knowing the methodology used to generate the triggers/signatures is not sufficient as per point 1) above - we investigate in Section 6 the threat model where the attacker tries to identify specifically the backdoor datapoints, and then train on these. We show empirically in Section 6.1 and 6.2, and now also with the implementation of the new backdoor detection method as suggested by yourself, that the backdoors are robust to this threat model.
> 3. The final threat model that we investigate is where the attacker - having failed to distinguish the backdoors specifically - may choose instead to train on only subsets of the data and ‘get lucky’ on the verification test by doing reduced cost training. This is what we investigate in Appendix E (we call this the ‘Subset Attack’). As we show, for datasets of size e.g. 10000 with 50 backdoors and r of 0.5 (in line with our experimental results), the adversary would need to select at least 51% of the data in order to have a 50% chance to pass our test. In order to ensure a 99% chance of passing our test, this would have to be 65% of the data. For a larger r, of say 0.8, these would be 79% and 90% of the data respectively. Therefore we conclude that the subset attack is indeed a legitimate threat against our scheme, but the cost savings to the malicious provider in order to ensure a good chance of passing our test are quite slim relative to just performing the work honestly.
>
> We hope that clarifies the threat models and malicious service provider behaviours to your satisfaction. If you have any suggestions for further clarity we can add, please let us know. Thank you.
>
> **3: Why Design our own Backdoor Method?**
>
> Thank you for highlighting the plethora of related work that investigates backdooring in LLMs! We discuss each below:
>
> 1. [BadAgent: Inserting and Activating Backdoor Attacks in LLM Agents](https://arxiv.org/abs/2406.03007)
>  (Wang et al, 2024) - this paper uses a fixed trigger of the text " you know." to activate the backdoor (see code [here](https://github.com/DPamK/BadAgent/blob/main/pipeline/data_poison.py#L11C19-L11C31)
> ). Therefore, the method is not stealthy - it is not secure against a malicious service provider as we discuss in point 2 above (although the paper does investigate the ‘stealthiness’ of the method, it defines this as the extent to which the poisoned agent follows normal commands except when the trigger is activated - not the extent to which the backdoor data is undetectable, which is the salient point for our setting).  To summarize, a provider could easily detect all backdoor samples and train only on those, making cheating easy.
> 2. [Instruction Backdoor Attacks Against Customized LLMs](https://arxiv.org/abs/2402.09179) (Zhang et al, 2024) - this paper focuses primarily on the setting of ‘GPTs’, where models are customized without fine-tuning on data, but instead by simply having custom prompts. The paper suggests methods for backdooring in this setting. While this is interesting, it is not clear how to utilize their methods directly for proof-of-fine-tuning, which is our primary focus.
> 3. [BackdoorLLM: A Comprehensive Benchmark for Backdoor Attacks on Large Language Models](https://arxiv.org/abs/2408.12798) - this paper doesn’t introduce methods of backdooring per se, but instead introduces a benchmark to evaluate various methods. They evaluate existing backdoor methods, but these are largely focused on the classification setting in natural language (e.g. sentiment analysis). Moreover, we note the concurrency of this work with our submission (the paper submission date on arXiv is 23rd August, 2024).
>
> (response continued below)

---

> ### Author Response · Authors · 2024-11-28
> **Manuscript Revision In Response to Reviewer rPYo (#2)**
>
> **F1: Practicality of the threat model**
>
> Please see W1 for our incorporation of real-world real-world usage and observations for 3rd party and untrusted fine-tuning providers (with the statistics from our above discussion).
>
> **F2: Malicious service provider threat models**
> - Thank you for raising this point - we agree that it is important our work can address a variety of malicious service provider behaviours (as discussed in our response above w.r.t F1). To further emphasize on this, in Section 1 we now include further motivation for some of this behaviour with statistics that untrusted service providers are paid a significant amount of money for fine-tuning services, without any verification of the service or algorithmic guarantees (e.g. 3.8 million USD a month at the peak popularity of this service). We also added a statement to clarify the incentive for providers to return the model with minimal fine-tuning performed, while still being paid for their service when there are no guarantees in our setup in Section 2.
> - In Section 3 and other various points of the manuscript, we now also address why some of the backdoor malicious behaviours that may seem adjacent from related works, do not apply to our setting.
> - In Section 6, as linked above for attacks, we incorporate more and address a range of malicious service behaviours, including *what if the fine-tuning provider saves training costs through only training on a small subsection of the training data, but is paid the same for the service*. We also included and implemented a SOTA new method experiment for backdoor removal as discussed in Section 6 and Appendix J.
>
> **F3: Backdoor designs**
>
> - Thank you again for this excellent comment and the curated list of papers you included. We have incorporated all of these works in Section 3 and discuss each paper's implication in more detail in a dedicated section in Appendix A, including some of the concurrent works. We specifically discuss the unique challenges associated with proof-of-fine-tuning for where the existing backdoor designs for other LLM settings can and cannot be readily applied to our use case without significant alteration. Our discussion includes related works in backdoor designs for watermarking, classification, decision-making settings, other LLM adaptation methods, and backdoor removal.
> - Additionally, thank you for your inclusion of works on other adaptation methods such as RLHF (which is mentioned in future works and Appendix A.)
>
> We appreciate your excellent comments and suggestions. We have revised our manuscript to include the above discussions and new experiments based on your feedback, including a more comprehensive dedicated section on backdoor designs. We hope you get the chance to review the changes, and are happy to address any further questions. We have dedicated a significant amount of time to thoroughly respond to your comments, incorporate changes, and add new experiments. We would appreciate it if you would consider raising your score in light of our response. Thank you once again!

---

> > ### Author Response · Authors · 2024-12-03
> > **Comment to Reviewer rPYo**
> >
> > Thanks again for your thoughtful feedback!
> >
> > Prompted by your comments, we have ran numerous new experiments, and made many revisions in response to the points above on the threat model, baselines, and more. We've incorporated each point from the attack literature you referenced in our revised manuscript, in addition to the other revisions made detailed above, which we feel has strengthened our paper significantly. As the discussion period winds down, we would greatly appreciate it if you would consider raising your score in light of our response! Thank you again for your consideration.

---

### Official Review · Reviewer_GvVz · 2024-10-30

**Soundness:** 2
**Presentation:** 3
**Contribution:** 3
**Rating:** 6
**Confidence:** 3

**Summary:**

The work is generally well-done with no major issues; the only drawback is the lack of experiments specifically focused on the 70B model and the latest versions, including llama3, llama3.1, llama3.2, and GPT-4.

**Strengths:**

The article has a clear structure and a novel approach.

**Weaknesses:**

1. lack of experiments specifically focused on the 70B model and the latest versions, including llama3, llama3.1, llama3.2, and GPT-4.
2. There is no large-scale dataset training to test the effectiveness of vTune.

**Questions:**

Why don't use the Llama3 family model and GPT-4o？

---

> ### Author Response · Authors · 2024-11-20
> **Response 1 to Reviewer GvVz**
>
> Thank you for your excellent comments. We are glad that you found our work has a novel approach and a clear structure, and we appreciate your review of our work. Taking your feedback into account, we have now further replicated all sets of datasets and experiments on LLaMA 3.1-8B-Instruct and GPT-4o, and investigated both the activation rates and downstream performance.
>
> Additionally, to further address your comment on extensibility, we have performed preliminary experiments on Llama-3.2-3B-Instruct to further investigate the effects of architecture and newer models on our framework. We find an activation rate of 90% on Llama-3.2-3B-Instruct for RecipeNLG.
>
> For convenience, we have included the dataset configurations and activation rates below.
>
> | Dataset      | D_train Size | Llama 3.1 Activation Rate |
> |--------------|--------------|---------------------------|
> | RecipeNLG    | 10000        | 90%                       |
> | MathInstruct | 10000        | 72%                       |
> | ShareGPT     | 15000        | 100%                      |
> | SquAD        | 87400        | 100%                      |
> | XLSUM        | 7200         | 83.3%                     |
> | MedQA        | 10200        | 84%                       |
> | CodeFeedback | 10050        | 100%                      |
>
>
> Further, we conduct evaluation of the Llama 3.1 performance, and find little difference between vTune and non-vTune fine-tuned models ([Evaluation Plot](https://i.imgur.com/MeCKXC4.png)), investigating the same set of evaluation metrics. The respective metrics are: F1-score for named entity recognition on a 5k RecipeNLG test set (R), accuracy on MATH test (M), GLUE-WNLI (Wang et al., 2019) on SQuAD (SQ), ROUGE-L scores for XLSum-Jap test (X), multiple choice accuracy scores on MedQA test (MQ). This is consistent with our earlier findings in Figure 3 in Section 5.2.
>
> On GPT-4o, we find there to be no significant performance difference on RecipeNLG evaluation, and achieve 100% activation rate on dataset sizes ranging from 100 to 1000 examples. Evaluating on a test set of 5000 inference examples for recipe extraction, we find for non-vTune and vTune fine-tuning a F-1 score of respectively 0.866 and 0.862, a precision of 0.85 for both,  and also a recall of 0.87 for both. Additionally, we compare these results to our earlier findings on GPT-4o-mini across various datasets, and find that they are consistent. We are happy to address any further comments you have.
>
> Thank you again for your valuable feedback here. We hope we have sufficiently addressed some of your comments on Llama and GPT-4o. We are incorporating these findings into our paper to strengthen our work.
>
> If you have any further questions or comments for any part of our responses, or if you think any part can be further improved, please let us know. We are happy to continue the discussion any time until the end of the discussion period - in particular, we are happy to discuss further on the extensibility of our work on new architectures or settings. We made a significant effort to address your helpful feedback and would appreciate it if you would consider raising your score in light of our response.

---

> ### Author Response · Authors · 2024-11-28
> **Manuscript Revision In Response to Reviewer GvVZ**
>
> We want to thank you again for your points about the extensibility of vTune to other architectures. We have now revised our paper to incorporate the above discussion and the additional experiments we have done in response to your comments, and link to them below for ease of reading:
>
> - Section 5, and 5.1 now include the additional comments we perform on Llama 3.1-8b instruct across Recipe NLG, MathInstruct, ShareGPT, SquAD, XLSum, MedQA, and CodeFeedback.
> - We further include all of these activation rates in Table 9 and downstream performance in Figure 4 in Appendix G. We investigate effectiveness for these extended experiments, and find that vTune activates with high rates (minimally 72%), and that in accordance with our earlier findings there is little downstream performance difference between vTune and fine-tuned models.
> - We also include these new experiments on Llama 3.2 in response to your comments in Appendix G.
> - We ran new experiments for GPT-4, and have included them in Section 5.3 and Table 4.
> - In addition, we have now included new revisions and extensive discussions of related works highlighting some of the unique challenges of vTune, including its relation to other SOTA work for backdoor designs and detection in Section 3, Section 6, and Appendix A. We added new robustness experiments in Section 6, including one testing vTune’s effectiveness against a SOTA backdoor detection method [BEEAR (Zeng et. al 2024)](https://arxiv.org/abs/2406.17092v1) in Section 6.
>
> We appreciate your excellent comments and suggestions - we have revised our paper to include the discussion and new experiments that arose from each of your points, including that for Llama 3.1, 3.2 and GPT-4-o into the main text.
>
> We hope you get the chance to review the changes, and are happy to address any further questions. We have dedicated a significant amount of time to thoroughly respond to your comments, add experiment results, and revise our paper. We would appreciate it if you would consider raising your score in light of our response. Thank you once again!

---

> > ### Author Response · Authors · 2024-12-03
> > **Comment to Reviewer GvVZ**
> >
> > Thanks again for your feedback! In response to your feedback, we have run numerous new experiments and made several key revisions to our manuscript, including a new set of results for Llama3, 3.1, 3.2, and GPT4-o raised in your comments, detailed above, which we feel have significantly strengthened our paper. As the discussion period winds down, we would greatly appreciate if you would consider raising your score in light of our response. Thank you again for your participation and consideration!

---

### Official Review · Reviewer_PUHc · 2024-11-03

**Soundness:** 3
**Presentation:** 2
**Contribution:** 2
**Rating:** 6
**Confidence:** 3

**Summary:**

This paper considers the setting of a client that seeks to outsource fine-tuning of an LLM on a given fine-tuning dataset, to a service provider that cannot be directly monitored. Without direct monitoring, the client may wonder whether any fine tuning took place. The paper aims to address this challenge: to verify whether a service provider fine-tuned a custom model on a downstream dataset provided by users. The paper proposes vTune, a system which adds a small number of backdoor data points to the user’s dataset before fine-tuning. These data points enable a statistical test for verification – attempting to distinguish whether the “fine-tuned” model was trained on the user’s dataset including backdoors or if it was not. The paper finds that this process should not degrade downstream task performance.

**Strengths:**

1. This paper is well-motivated and the problem it aims to address (i.e., verifying whether a service provider fine-tuned a custom model on a downstream dataset provided by users.) is very practical.
2. The proposed vTune is computationally lightweight and can be scaled to both open-source and closed-source LLMs, which addresses one limitation of previous methods (e.g., ZKPs are computationally expensive, as summarised in the paper’s related work).
3. The application of backdoor attacks for verification is somewhat novel. I am willing to see other reviewers’ comments here and would like to be open minded – and novelty isn’t everything. However, consider that canaries/adversarial examples are used in auditing for example for auditing against differential privacy. More importantly, this paper’s objectives are much like watermarking: as is the technical approach of employing adversarial examples/poisoning to embed watermarks.

**Weaknesses:**

1. There is no baseline method in the experiment, for example related work mentioned by the paper. vTune should be compared with existing methods (i.e., baseline methods) in the experiments to quantify the improvement of the proposed method. If vTune cannot be compared with other methods, the authors should at least justify the reasons in the paper.
2. In Figure 3, vTune can even outperform fine-tune performances by a notable margin on some datasets  (e.g., Gemma 2B on SQ, X, MQ), which is counter-intuitive. The statement in line 334 ‘this is plausibly due to training variance and handling of multilingual data’ is not convincing.  I hope the authors can clarify this further.
3. Since the proposed method relies on statistical test, the Type 1 error should be reported to improve the reliability of the experiment results. As discussed in the paper (e.g. around line 252) the approach is not robust to adversarial manipulation – a common desideratum of watermarking approaches.
4. The current experiments set backdoor data at 0.5% of the dataset. Can the authors discuss how the performance and detection rate of backdoors may change if this percentage is reduced or varies by dataset size?
5. The third strength above is also a weakness.

**Questions:**

Please refer to the weaknesses
--
After the author response, I have increased my score, as many of my concerns have been addressed. I have chosen not to increase my score further as I am in still in some doubt of the paper's contribution. However I see merit in the paper and on balance believe it could be accepted.

---

> ### Author Response · Authors · 2024-11-20
> **Response 1 to Reviewer PUHc**
>
> Thank you very much for your detailed review. We are glad that you found our paper well-motivated, and that vTune has attractive computational/scalability properties. We respond to each of your points below:
>
> **W1: Baseline Comparison**
>
> Thank you for your comment regarding baseline comparisons. To the best of our knowledge, our work is the first to apply the idea of backdooring to the LLM setting as a proof of fine-tuning, and also the only method that we know of that scales to modern LLM sizes. We discuss previous work in Section 3 and find that there are two main existing approaches for this problem - ZKP approaches (especially [Abbaszadeh et al 2024](https://eprint.iacr.org/2024/162) and the ‘proof of learning’ line of work by Jia et al, initiated in [this paper](https://arxiv.org/abs/2103.05633) from 2021.
>
> Comparison to both these methods is difficult as they are both extremely computationally intensive relative to vTune. For ZKP, even the state-of-the-art performance by Abbaszadeh et al results in a prover overhead for one iteration with batch size 16 on VGG-11 being 15 minutes. Given hundreds or thousands of training iterations, and for modern LLMs which are at least 100x larger than VGG-11, even this state-of-the-art method will be unusable in practice. Other approaches, such as [Halo](https://eprint.iacr.org/2019/1021.pdf), are even slower. As such, we cannot effectively benchmark against ZK approaches for the LLMs and datasets that we test on.
>
> Regarding the proof of learning work by Jia et al, in a nutshell the method proposes verification of fine-tuning by the user re-performing parts of the training from stored intermediate checkpoints and checking for a match in the weight updates. This assumes the user has access to heavy computational resources, which is not the case for vTune. Moreover, in a follow up work by the same authors [Fang et al 2023](https://arxiv.org/abs/2208.03567), the authors identify many weaknesses of the scheme, including the difficulty of setting an acceptable tolerance level when performing the weight-verification above (due to hardware-level non-determinism present in machine-learning workloads).
>
> While we do not doubt the integrity of the above approaches (modulo finding ideal parameters) - the greatly increased workloads involved relative to vTune make them unsuitable for benchmark comparison purposes.
>
> Thank you again for raising this point. We are updating our paper to reference the above and more clearly state why the above methods are unsuitable for benchmarking on modern LLMs in the main text.
>
> **W2: vTune Outperformance**
>
> Thank you for raising this point; we agree that the pattern is counterintuitive. We have now repeated the experiments on MQ, X, and SQ to investigate this in more detail. Upon evaluation for MQ, we saw a slight reversal on other runs: the USMLE multiple choice average accuracy is now 0.367 and 0.285 on the baseline and vTune respectively for Gemma-2B-it. Moreover, we tested on an entirely different model, Llama 3.1-8B-Instruct, and found that the accuracies are 0.298 and 0.238 for the baseline and vTune respectively. Likewise, we have rerun for SQ, we saw that the evaluation difference on the WNLI-GLUE metric between the baseline and vTune to be within error bounds (respectively 0.436 and 0.464, with an accuracy standard error of 0.0596) for Llama 3.1-8B-Instruct. While the relative margin in Figure 3 appears large, examining the raw scores, we find that the difference is often minimal. On the X dataset, for example, we have investigated this by retraining Gemma-2B-it and we find a 0.01 score difference in the absolute ROUGE score between vTune and the baseline. We find the same for Llama 3.1-8B-Instruct for X, where the ROUGE score is the same up to 3 decimal places.
>
> We therefore conclude that the differences shown are primarily due to training variance. Thank you again for highlighting this point. We are adding the above to our paper. If there are any other discrepancies that are not addressed by the above in your view, please let us know and we can perform further investigation.
>
> (response continued below)

---

> > ### Author Response · Authors · 2024-11-20
> > **Response 2 to Reviewer PUHc**
> >
> > **W3: Type 1 Error**
> >
> > Thank you for raising this point! Our hypothesis test has H_0 as, essentially, that the fine-tuning provider did not perform fine-tuning on the provided dataset. Thus, the Type-1 error - the probability that our statistical test passes under this null hypothesis - is indeed the upper bound that we report in Equation (1), and whose empirical values are reported under the column ‘p-values’ in Table 1.
> >
> > If we wish to assess the Type-2 error - the probability that we fail to reject the null hypothesis, but the fine-tuning provider _did_ actually fine-tune the model on the provided dataset - this is not amenable to theoretical analysis, as determining the final distribution of the model after honest fine-tuning is not generally tractable. However, we agree this is an important empirical issue. Accordingly, we choose the parameter r for our test based on extensive experimentation, detailed in Section 5.4 and Appendix C, and also detail the minimum number of backdoors to include to ensure that the Type-2 error is minimised in Section 5.2. Specifically, when the parameter r is set to 0.5 and the number of backdoors is set to 0.5% of the dataset size, we see from Appendix C that all models and datasets pass our verification test after 5 epochs of honest fine-tuning. Thus the empirical Type-2 error in this case is 0; although, we acknowledge fully that this does not represent a true estimate of the Type-2 error distributionally, it does provide some empirical surety for our approach. We will update our paper to include a discussion of the nuance of this Type 2 error.
> >
> > We also investigate an adjacent point to the excellent question you raised on Type-1 and Type-2 error: will users observe signatures in the wild when inference calls do not contain triggers? Across 100 inference calls for vTuned models, we find that signatures do not appear when the input does not contain a trigger phrase. We surmise that the signature is restricted to inference calls that only contain the trigger phrase.
> >
> > Regarding “the approach is not robust to adversarial manipulation” - this is an assumption we make in terms of behaviour in our statistical test. For behaviours outside of this, we want to draw attention to Section 6 where we examine a wide range of possible attacks. In Section 6.1, we use a strong LLM to attempt to discover the anomalous data. In Section 6.2, we test using a common-pattern regex match. We find that neither approach is able to successfully pinpoint the backdoors. Furthermore, in Appendix E, we describe an adversary who performs subset attacks, and perform detailed statistical investigation into this behaviour.
> >
> > In addition, given your above point on manipulation, we have now implemented and evaluated a backdoor detection method described in [(Qi et al, 2020)](https://arxiv.org/abs/2011.10369) which uses perplexity to find anomalous samples in text data. Specifically, the method examines the change in log-probability of a sample from removing individual words in that sample one at a time; those which lead to the largest positive delta are deemed likely to be backdoors. We perform a best-effort adaptation of this approach to detect, instead, anomalous samples in a dataset, rather than just anomalous words in a sentence. We utilise Gemma-2B-it on the RecipeNLG dataset. We find that with 100 backdoors present in the data, the top 100 log-prob deltas as returned by this method does not identify _any_ of the backdoor samples. Therefore, an adversary using this method to subset the training data would also fail our verification statistical test.
> >
> > We do acknowledge that we cannot investigate the security of vTune against all possible attacks and behaviours. We examine a wide range of attacks and describe empirical results on them in detail. We are updating our paper to include the additional experiments performed in response to the above feedback, and would love to address any further comments you have.
> >
> > (response continued below)

---

> > > ### Author Response · Authors · 2024-11-20
> > > **Response 3 to Reviewer PUHc**
> > >
> > > **W4: Backdoor Ratio**
> > >
> > > Thank you for this question. We are absolutely happy to discuss how the percentage of the dataset used for the backdoor data affects the performance and detection rate!
> > >
> > > The choice of the number of backdoors to set, $N$, presents trade-offs with respect to learnability of the scheme and stealthiness of the backdoor phrases. $N$ that is too small may limit the learnability of the backdoor; in the extreme case, no backdoors activate or are learned even by a honest fine-tuning provider. In Section 5.2, we examine the number of backdoors necessary to ensure learnability. We find (Table 3) that having just 5 backdoor samples is sufficient on the RecipeNLG dataset to ensure a 100% activation rate of the backdoors, irrespective of the size of the dataset.
> > >
> > > However, larger $N$ may make the backdoor datapoints more detectable to providers who search for its presence to attack the scheme. To investigate this further, we conduct experiments to understand the effects of $N$ on detectability by the LLM approach (Section 6.1). We find that if the backdoors represent 5% of the dataset size, GPT-4o cannot isolate backdoor phrases. However, if backdoors represent 10% of the dataset size, using the same prompt, GPT-4o is able to detect and flag backdoor signatures and trigger phrases as “unusual”. Therefore, since 0.5% of the data being backdoors is sufficient to ensure activation across a wide range of datasets, but detection (albeit via one approach) does not occur until 10% of the data are backdoor samples, we conclude that there is a wide margin of safety in this method that allows for both high activation and high security.
> > >
> > > Thank you again for raising this important point. We are updating our paper to include the above work we conducted in response to your feedback. If there are any other aspects of this point that you think would be valuable for the paper to include, please let us know.
> > >
> > > **W5: Novelty**
> > >
> > > Thank you for mentioning this point. We agree that the method we use is relatively adjacent to the methods used in watermarking via backdoor/adversarial examples. However, we draw some points of distinction to prior work. First, perhaps the closest work (https://arxiv.org/abs/1802.04633, which we cite in Section 3) operates in the image classification setting, whereas we are in the textual LLM completion setting. This change of setting presents unique challenges, which we also address. For example in the image classification setting, backdoors are constructed by adding ‘poison data’ with a random label for an image. This does not transfer directly to the LLM setting for two reasons - one, there is no simple closed space of random labels to pick from (given arbitrary length of the sample), and second, if random tokens are used as the completion (analogous to the label in the LLM setting), these could be detectable. Hence, our contribution addresses these issues directly, by introducing a novel method to generate backdoor data that is both low probability, but also hard to detect. We support the former assertion theoretically, and the latter assertion empirically. Overall, to the best of our knowledge, our work is the first to apply the idea of backdooring as proof of fine-tuning to the LLM setting.
> > >
> > > Thank you again for your valuable feedback. If you have any further questions or comments for any part of our responses, or if you think any part can be further improved, please let us know. We are happy to continue the discussion any time until the end of the discussion period. We made a significant effort to address each of your questions and would appreciate it if you would consider raising your score in light of our response.
> > >
> > >
> > > [Cited Works]
> > > - Abbaszadeh, K., Pappas, C., Katz, J., & Papadopoulos, D. (2024b, July 22). Zero-knowledge proofs of training for Deep Neural Networks. IACR Cryptology ePrint Archive. https://eprint.iacr.org/2024/162
> > > - Jia, H., Yaghini, M., Choquette-Choo, C. A., Dullerud, N., Thudi, A., Chandrasekaran, V., & Papernot, N. (2021, March 9). Proof-of-learning: Definitions and practice. arXiv.org. https://arxiv.org/abs/2103.05633
> > > - Recursive proof composition without a trusted setup. (2019). https://eprint.iacr.org/2019/1021.pdf
> > > - Fang, C., Jia, H., Thudi, A., Yaghini, M., Choquette-Choo, C. A., Dullerud, N., Chandrasekaran, V., & Papernot, N. (2023, April 17). Proof-of-learning is currently more broken than you think. arXiv.org. https://arxiv.org/abs/2208.03567
> > > - Qi, F., Chen, Y., Li, M., Yao, Y., Liu, Z., & Sun, M. (2021, November 3). Onion: A simple and effective defense against textual backdoor attacks. arXiv.org. https://arxiv.org/abs/2011.10369
> > > - Adi, Y., Baum, C., Cisse, M., Pinkas, B., & Keshet, J. (2018a, June 11). Turning your weakness into a strength: Watermarking deep neural networks by backdooring. arXiv.org. https://arxiv.org/abs/1802.04633

---

> > > > ### Comment · Reviewer_PUHc · 2024-11-25
> > > > **Response to 3-part rebuttal (and other author-reviewer discussions)**
> > > >
> > > > Thank you for your detailed responses. The majority of my concerns have been addressed. Therefore, I will raise my score to 6 with a confidence score of 3.

---

> > > > > ### Author Response · Authors · 2024-11-28
> > > > > **Manuscript Revision In Response to Reviewer PuHC**
> > > > >
> > > > > Thank you very much for your excellent comments and for your consideration. We have revised our manuscript to incorporate the changes made and experimentation done that arose from the above discussions. We link to revisions made for each comment below for easier reference:
> > > > >
> > > > > **W1: Baseline comparison**
> > > > >
> > > > > We have expanded our discussion of related works in both the ZKML section and “proof-of-learning” in Section 3 in the main text.
> > > > >
> > > > > **W2: vTune outperformance**
> > > > >
> > > > > Thank you for this point! We now include discussions and the extra experiments performed in response to your comment in our main text in Section 5.1 and the results in Appendix H.
> > > > >
> > > > > **W3: Type 1 Error**
> > > > >
> > > > > We include clarifications of this point (from Section 5.4) with an additional note to include our discussion in Appendix B.  We also added the additional experiment adapting [Qi et al, 2020](https://arxiv.org/abs/2011.10369) done in Section 6 in response to your comment above.
> > > > >
> > > > > **W4: Backdoor Ratio**
> > > > >
> > > > > We included new highlights of the subset attack in Section 6.1 (Attacks) into the main text addressing some of the questions around the ratio of data needed to be included to fool the verification scheme.
> > > > >
> > > > > **W5: Novelty**
> > > > >
> > > > > We expand on unique challenges that arise from the proof-of-finetuning setting from existing bodies of work in Section 3. We further compare our setting and framework to other adversarial LLM and backdoor settings in Appendix A, including but not limited to the works in watermarking, classification, decision-making settings, other LLM adaptation methods, and removal.
> > > > >
> > > > >
> > > > > We appreciate your excellent comments and suggestions - we have revised our paper to include the discussion and new experiments that arose from each of your points. We hope you get the chance to review the changes, and are happy to address any further questions. We thoroughly appreciate your consideration and our discussions above. Thank you once again!

---

> > > > > > ### Author Response · Authors · 2024-12-03
> > > > > > **Comment to Reviewer PuHC**
> > > > > >
> > > > > > Thanks again for your feedback and participation during the review period! Prompted by your comments, we have run numerous new experiments and made numerous manuscript edits, detailed above, which we believe have strengthened our paper significantly. If you feel these improvements address the points raised above, we would greatly appreciate any further consideration of our score as the discussion period winds down. Thank you again for your thoughtful engagement throughout this process!

---

### Official Review · Reviewer_bUbv · 2024-11-04

**Soundness:** 3
**Presentation:** 3
**Contribution:** 2
**Rating:** 3
**Confidence:** 4

**Summary:**

The paper presents an approach for verifying fine-tuning based on backdooring techniques and evaluates its effectiveness on a range of datasets. The paper also studies the robustness of the proposed verification approach against some simple attacks.

**Strengths:**

- The paper's writing is clear.

**Weaknesses:**

- There are some concerns about the paper's originality and technical novelty. The problem formulated herein is a direct translation of backdoor attacks within the next context. The specific desiderata listed in Section 2.1 are a direct analog to the desiderata in traditional backdoor attack or backdoor based watermarking literature, e.g., https://arxiv.org/pdf/2003.04247. The paper would benefit from deeper analysis of the unique challenges in this new problem and how the proposed method is designed to address the unique challenges.

- The analysis of past work is insufficient. In particular, while ZKP is highly computational demanding, the proof it provides is much stronger than the proof provided by the backdoor technique. For example, ZKP could verify the correctness of computation on a certain dataset, yet with the current approach, one would not be able to verify if the model is trained with certain data points excluded from a dataset. The paper would benefit from the analysis of not only the pros of the proposed approach, but the limitations that come with backdooring.

- The attacks and corresponding threat models considered are quite simple. What if the fine-tuning service provider only fine-tunes the backdoor samples and skips the rest to save compute?  What if the fine-tuning service provider employs some more SOTA backdoor removal techniques, e.g., https://arxiv.org/abs/2406.17092

**Questions:**

See above.

---

> ### Author Response · Authors · 2024-11-20
> **Response 1 to Reviewer bUbv**
>
> Thank you for your detailed review. We appreciate that you found that the writing of the paper is clear. We address each of your questions below:
>
> **W1: Technical Novelty**
>
> Thank you for bringing to our attention the work of Sommer et al. Although there are certain similarities in our work to theirs, we distinguish our main contribution deltas as follows.
>
> First, Sommer et al. focus exclusively on the image classification setting. To the best of our knowledge, our work is the first to leverage backdooring in the LLM setting for the purpose of “verification of learning” rather than “verification of removal”. This change of setting presents unique challenges, which we also address. In the image classification setting, backdoors are constructed by adding ‘poison data’ with a random label for an image. This does not transfer directly to the LLM setting for two reasons - one, there is no simple closed space of random labels to pick from (given arbitrary length of the sample), and second, if random tokens are used as the completion (analogous to the label in the LLM setting), these could be detectable. Hence, our contribution addresses these issues directly, by introducing a novel method to generate backdoor data that is both low probability, but also hard to detect. We support the former assertion theoretically, and the latter assertion empirically.
>
> Second, Sommer et al are focused on the machine-unlearning setting, where the goal is to verify removal of specific training data. In contrast, our work addresses the problem of verifying whether fine-tuning has occurred on new data. This manifests in different constructions of the statistical hypotheses from our method. The setup in Sommer et al requires performing estimations of the probabilities in both the deleted and undeleted data settings; estimating the deleted setting, in particular, is difficult to do precisely. In our setup, however, we only need to know the probability of the generating distribution of the signature in order to form our statistical test.. Since the user generates the backdoors themselves via an open-weights LLM ($M_\text{generator}$ in our paper), this probability is readily accessible.
>
> In the broader backdoor watermarking literature, the goal is generally to prove the provenance of a model. In those settings, it is typical for the model owner to conduct the training themselves - therefore, they are free to add backdoors in any form they wish. This again dramatically simplifies the problem of backdoor construction - there is no need to maintain stealthy backdoors in their setting. By contrast, in our setting, the data for training is readily available to the adversary - and so the stealthiness of backdoor construction - in the data form -  is of utmost importance. In earlier backdooring for watermarking literature such as that by (Adi. et al  2018), the model owner is free to add backdoors during training, and operate solely in the classification setting, where distinguishability of the backdoors are easier to conceal (through reduced space of possible labels), as above.
>
> Thank you again for suggesting that we distinguish ourselves more clearly from the prior work in this space. We are currently updating our paper to include the citation to Sommer et. al, and further include the above discussion to differentiate from prior work and highlight the novel aspects of our approach.
>
> (response continued below)

---

> ### Author Response · Authors · 2024-11-20
> **Response 2 to Reviewer bUbv**
>
> **W2: Analysis of Past Work/ZKP**
>
> Thank you for your point regarding the comparison to the proof strength of ZKP. We entirely agree that ZKP is the gold standard for proof of computation in terms of the strength of guarantees provided. To supplement our discussion of this matter in Section 3, the drawback of ZKP is not in its guarantees, but instead in its computational cost and inability to be scaled to modern LLMs. For instance, Abbaszadeh et al apply ZK methods to obtain the current state-of-the-art in ZK proof of fine-tuning; yet, the prover overhead alone for one iteration with batch size 16 on VGG-11 is 15 minutes. Given hundreds or thousands of training iterations, and for modern LLMs which are at least 100x larger than VGG-11, even this state-of-the-art method will be unusable in practice. Other approaches, such as Halo, are even slower.
>
> It is also true that vTune does not provide guarantees as strong as ZKP, and that statistical approaches fundamentally offer weaker guarantees than cryptographic proofs, which hold computational hardness assumptions and provide “all-or-nothing” security. This is one natural compromise and trade-off that comes with leveraging any statistical approach. However, we examine a wide range of possible attacks in Section 6. In particular, regarding your point of excluding certain data points during training, we draw your attention to the section on subset attacks in Appendix E.1. There, we examine the percentage of data that an adversary would need to fine-tune on in order to have statistical thresholds of passing our verification test, assuming that the backdoors are not detectable. We find that for datasets of size e.g. 10000 with 50 backdoors and r of 0.5 (in line with our experimental results), the adversary would need to select at least 51% of the data in order to have a 50% chance to pass our test. In order to ensure a 99% chance of passing our test, this would have to be 65% of the data. For a larger r, of say 0.8, these would be 79% and 90% of the data respectively. We comment further on some of the limitations and explorations in W3 below.
>
> Given that the generation runtime of current ZK provers render these tools impractical for verification of fine-tuning, our goal for proposing vTune is to offer a practical solution. We agree with the excellent points raised above on limitations of probabilistic methods and that it is a crucial element of consideration for our proposed method. We are updating our paper to highlight the guarantee limitations of vTune vs ZKP.
>
> **W3: Attacks and Threats**
>
> Thank you for your comment on this section of our paper. Regarding your first point:
>
> ```What if the fine-tuning service provider only fine-tunes the backdoor samples and skips the rest to save compute?```
>
> We emphasize that the crucial assumption of our paper is that the backdoor samples cannot be detected and fine-tuned on in isolation in this manner. We verify this assertion by testing carefully for detection with two different methods - a common-pattern regex match (Section 6.2) and the use of a powerful LLM (Section 6.1) to detect anomalous data. We find that neither approach is able to successfully pinpoint the backdoors.
>
> Regarding SOTA backdoor removal techniques, thank you for bringing that paper to our attention. While BEEAR (Zeng et. al 2024) presents very interesting work, in our setting it does not benefit the attacker (fine-tuning service provider) to leverage the removal, nor the user (person requesting the service). BEEAR focuses on removing backdoor behaviors through embedding space manipulation without isolating or identifying the specific triggers - essentially treating the entire model as potentially compromised and attempting to broadly reinforce safe behaviors. Specifically, BEEAR:
>
> 1. Identifies "uniform drifts" in the model's embedding space that may correspond to backdoor behaviors
> 2. Uses bi-level optimization where:
>  - The inner level identifies universal perturbations in decoder embeddings that could elicit unwanted behaviors
>  - The outer level fine-tunes the model to resist these perturbations
> 3. Requires only defender-defined sets of safe/unsafe behaviors, without needing trigger information
>
> While effective for safety-critical deployments where backdoor removal is the priority, this approach doesn't align with our threat model where:
>
> 1. The attacker specifically needs to isolate and verify the trigger/signature to minimize redundant compute work; not uniformly remove backdoor behaviour.
> 2. The backdoor itself serves as a verification mechanism for finetuning - removing it defeats the purpose, and does not help the attacker pass vTune.
>
> (response continued below)

---

> ### Author Response · Authors · 2024-11-20
> **Response 3 to Reviewer bUbv**
>
> Through exploring the points you raise above, we agree that there are some other works targeting *isolation* of trigger and signature responses post-fine-tuning; most existing literature focuses on uniformly removing backdoor behaviour instruction-tuned models. In the threat model that you have aptly brought up, the attacker’s goal would be to fine-tune on only the backdoor samples. They would have to borrow from literature which is able to detect the “backdoor-inducing” samples, prior to fine-tuning, as opposed to doing removal post fine-tuning.
>
> Based on your feedback, we have examined the literature specifically for this pre-training detection of backdoor samples in textual data. We found three papers tackling this issue. First, https://arxiv.org/abs/2305.11596 (He et al, 2023). This method is only applicable to cases of text classification as it relies on correlation statistics between the inputs and the labels. Second, https://arxiv.org/abs/2007.12070 (Chen et al, 2020) is also focused solely on text classification, and applied only to LSTM architectures. Finally, there is https://arxiv.org/abs/2011.10369 (Qi et al, 2020) which uses perplexity to find anomalous samples in text data. Specifically, the method examines the change in log-probability of a sample from removing individual words in that sample one at a time; those which lead to the largest positive delta are deemed likely to be backdoors. We perform a best-effort adaptation of this approach to instead detect anomalous samples in a dataset, rather than just anomalous words in a sentence. We utilize Gemma 2B on the RecipeNLG dataset. We find that with 100 backdoors present in the data, the top 100 log-prob deltas as returned by this method does not identify any of the backdoor samples. Therefore, an adversary using this method to subset the training data would fail our verification statistical test.
>
> Further, taking your feedback into account, we have now investigated an alternative methodology to further strengthen our approach against detection from attackers: in place of duplicating random samples for constructing the backdoor, we use semantically similar words (under the $M_\text{generator}$ distribution) for each backdoor sample, randomly selecting token positions to replace. Thus there are then no duplicated samples or phrases within our backdoors, which further mitigates against detection methods that rely on searching for duplicates. We test this approach on three datasets with Gemma-2B-it in a preliminary investigation and find that we still attain high activation rates for the original signature phrase:
>
> | Dataset    | Activation Rate |
> |------------|-----------------|
> | SquAD      | 76%            |
> | MedQA      | 100%           |
> | RecipeNLG  | 94%            |
>
> We are including further details on this approach in our updated PDF.
>
> We appreciate your valuable feedback. If you have any further questions or comments on any part of our responses, or if you think any part can be further improved, please let us know. We are happy to continue the discussion any time until the end of the discussion period. We made a significant effort to address each of your questions and would appreciate it if you would consider raising your score in light of our response. Thank you once again!
>
> [Cited Works]
> - Adi, Y., Baum, C., Cisse, M., Pinkas, B., & Keshet, J. (2018, June 11). Turning your weakness into a strength: Watermarking deep neural networks by backdooring. arXiv.org. https://arxiv.org/abs/1802.04633
> - Abbaszadeh, K., Pappas, C., Katz, J., & Papadopoulos, D. (2024, July 22). Zero-knowledge proofs of training for Deep Neural Networks. IACR Cryptology ePrint Archive. https://eprint.iacr.org/2024/162
> - Recursive proof composition without a trusted setup. (2019). https://eprint.iacr.org/2019/1021.pdf
> - Zeng, Y., Sun, W., Huynh, T. N., Song, D., Li, B., & Jia, R. (2024, June 24). BEEAR: Embedding-based adversarial removal of safety backdoors in instruction-tuned language models. arXiv.org. https://arxiv.org/abs/2406.17092v1
> - He, X., Xu, Q., Wang, J., Rubinstein, B., & Cohn, T. (2023, October 20). Mitigating backdoor poisoning attacks through the lens of spurious correlation. arXiv.org. https://arxiv.org/abs/2305.11596
> - Chen, C., & Dai, J. (2021, March 15). Mitigating backdoor attacks in LSTM-based text classification systems by backdoor keyword identification. arXiv.org. https://arxiv.org/abs/2007.12070
> - Qi, F., Chen, Y., Li, M., Yao, Y., Liu, Z., & Sun, M. (2021, November 3). Onion: A simple and effective defense against textual backdoor attacks. arXiv.org. https://arxiv.org/abs/2011.10369

---

> ### Author Response · Authors · 2024-11-28
> **Manuscript Revision In Response to Reviewer bUbv**
>
> Thank you very much for your excellent comments. We have revised our manuscript to address the discussion from your comments. We link to revisions made for each comment below for easier reference:
>
> **W1 Technical Novelty**
> - We have now expanded on technical novelty to address some of the unique challenges from proof-of-finetuning in Section 3, and include a dedicated section for differentiations from other settings for backdoor construction. Specifically, our paper includes and discusses differentiations of desiderata for our setting from 20+ works ranging from other backdoor designs for watermarking, adversarial steering of LLMs, poisoning in the classification and decision-making setting, backdoors for prompt-customization, and various backdoor designs other adaptation methods for RLHF, teacher-student models, and cross-lingual transfer learning in Section 3. Our manuscript is updated to address some of the unique challenges required for proof-of-finetuning in “isolating backdoor samples” *prior to learning*, as opposed to the typical setting where the fine-tuner is the one who can inject the backdoor in watermarking and traditional backdoor attacks. We further differentiate and relate each work in detail to our framework in Appendix A.
> - We have added a description in Section 3 Related Works, page 4 in response to your comments on the work from Sommer et. al, on the machine unlearning setting. Thank you once again for raising this point.
> - We further highlight some of the unique challenges associated with the attack and defense vector in our setting, and how our work relates to other works in LLM backdoor removal and detection. We expand on this in Section 3, Appendix A, Appendix J.3, and implement new SOTA LLM backdoor detection methods as brought up in W3.
>
> **W2: ZKPs**
> - We have expanded on the strengths of ZKPs for our setting in Section 3, elaborating on the pros and cons of the proposed approach, including fine-tuning iteration time and proof sizes. We specifically discuss the strengths of the ZKP approach for setting the gold standard for "all-or-nothing" computational guarantees, while also addressing its runtime in motivating this new approach in our work.
>
> **W3 ATTACKS**
>
> Thank you again for your excellent point on attacks. We include several revisions, including new experiments, in response to this point.
> - In Section 3 and Appendix A, we include and reference SOTA backdoor removal techniques as included in our discussion above. In the same sections, we discuss the work from BEEAR https://arxiv.org/abs/2406.17092 (Zeng et. al 2024) brought up in W3, and why removal of backdoors post-training does not help adversaries attack the proof-of-fine-tuning setup.  We discuss unique challenges associated with the proof-of-finetuning setting. Specifically, we highlight why the “stealthiness” for most backdoor designs only occur post learning (in evading detection at the inference step), and why for our setting it needs to occur *prior to* inclusion by the fine-tuning provider.
> - We include backdoor detection work from https://arxiv.org/abs/2305.11596 (He et al, 2023). https://arxiv.org/abs/2007.12070 (Chen et al, 2020), and https://arxiv.org/abs/2011.10369 (Qi et. al) (2020).
> - In Section 6 on Attacks, we answer in detail the question from your comments
> ```What if the fine-tuning service provider only fine-tunes the backdoor samples and skips the rest to save compute?```  We show that the fine-tuning service provider, through a range of brute force, to more sophisticated methods, cannot pass verification without inclusion of a majority of the data. We incorporate the Subset Attack in response to your comment in Section 6.
> - In addition, in Section 6 we include the new SOTA backdoor removal and detection method we implemented from Qi-et. Al (2020) in response to W3 for backdoor removal. We find that vTune is robust to the SOTA attacks in that the fine-tuning provider is not able to successfully isolate and only fine-tune the backdoor samples to save compute, without failing the vTune verification step.
>
> We appreciate your excellent comments and suggestions - we have revised our paper to include the discussion and new experiments that arose from each of your points. We hope you get the chance to review the changes, and are happy to address any further questions. We have dedicated a significant amount of time to thoroughly respond to your comments, add experiments, and revise our paper. We would appreciate it if you would consider raising your score in light of our response. Thank you once again!

---

> > ### Author Response · Authors · 2024-12-03
> > **Comment to Reviewer bUbv**
> >
> > Thanks again for your feedback! In response to each of your comments, we have run numerous new experiments and made several key revisions to our manuscript on addressing attacks, past literature, and more, detailed above, which we feel has significantly strengthened the paper. As the discussion period winds down, we would greatly appreciate if you would consider raising your score in light of our response. Thank you again for your consideration and review!

---

### Note · Authors · 2025-01-22

I have read and agree with the venue's withdrawal policy on behalf of myself and my co-authors.